# Cardiac mitochondrial function depends on BUD23 mediated ribosome programming

Matthew Baxter[1,2,3]*, Maria Voronkov[1,2,3], Toryn Poolman[1,2,3], Gina Galli[1], Christian Pinali[4], Laurence Goosey[1], Abigail Knight[1], Karolina Krakowiak[1], Robert Maidstone[1,2,3], Mudassar Iqbal[4], Min Zi[4], Sukhpal Prehar[4], Elizabeth J Cartwright[4], Julie Gibbs[1], Laura C Matthews[5], Antony D Adamson[1], Neil E Humphreys[1], Pedro Rebelo-Guiomar[6,7], Michal Minczuk[7], David A Bechtold[1], Andrew Loudon[1], David Ray[1,2,3]*

[1]Faculty of Biology, Medicine and Health, University of Manchester, Manchester Academic Health Science Centre, Manchester, United Kingdom; [2]Oxford Centre for Diabetes, Endocrinology and Metabolism, University of Oxford, Oxford, United Kingdom; [3]NIHR Oxford Biomedical Research Centre, John Radcliffe Hospital, Oxford, United Kingdom; [4]Division of Cardiovascular Sciences, University of Manchester, Manchester Academic Health Science Centre, Manchester, United Kingdom; [5]Leeds Institute of Medical Research, Faculty of Medicine and Health, University of Leeds, Leeds, United Kingdom; [6]Graduate Program in Areas of Basic and Applied Biology (GABBA), University of Porto, Porto, Portugal; [7]Medical Research Council Mitochondrial Biology Unit, University of Cambridge, Cambridge, United Kingdom

**\*For correspondence:**
matthew.baxter@ocdem.ox.ac.uk (MB);
david.ray@ocdem.ox.ac.uk (DR)

**Competing interests:** The authors declare that no competing interests exist.

**Abstract** Efficient mitochondrial function is required in tissues with high energy demand such as the heart, and mitochondrial dysfunction is associated with cardiovascular disease. Expression of mitochondrial proteins is tightly regulated in response to internal and external stimuli. Here we identify a novel mechanism regulating mitochondrial content and function, through BUD23-dependent ribosome generation. BUD23 was required for ribosome maturation, normal 18S/28S stoichiometry and modulated the translation of mitochondrial transcripts in human A549 cells. Deletion of *Bud23* in murine cardiomyocytes reduced mitochondrial content and function, leading to severe cardiomyopathy and death. We discovered that BUD23 selectively promotes ribosomal interaction with low GC-content 5'UTRs. Taken together we identify a critical role for BUD23 in bioenergetics gene expression, by promoting efficient translation of mRNA transcripts with low 5'UTR GC content. BUD23 emerges as essential to mouse development, and to postnatal cardiac function.

## Introduction

Regulation of protein abundance within a cell is of fundamental importance to homeostasis, cell identity and responsiveness to changes in external environment. Many layers of regulatory control have evolved to fine-tune gene expression to meet cellular needs, including transcriptional regulation, cis-acting genomic elements, epigenetic structures, and translational efficiency. Selective control of mRNA translation at the level of the ribosome is emerging as important for protein dynamics within the cell; however, the regulatory mechanisms remain unclear (*Sonenberg and Hinnebusch,*

**eLife digest** Cells need to make proteins to survive, so they have protein-making machines called ribosomes. Ribosomes are themselves made out of proteins and RNA (a molecule similar to DNA), and they are assembled by other proteins that bring ribosomal components together and modify them until the ribosomes are functional.

Mitochondria are compartments in the cell that are in charge of providing it with energy. To do this they require several proteins produced by the ribosomes. If not enough mitochondrial proteins are made, mitochondria cannot provide enough energy for the cell to survive.

One of the proteins involved in modifying ribosomes so they are functional is called BUD23. People with certain diseases, such as Williams-Beuren syndrome, do not make enough BUD23; but it was unknown what specific effects resulted from a loss of BUD23.

To answer this question, Baxter et al. first genetically removed BUD23 from human cells, and then checked what happened to protein production. They found that ribosomes in human cells with no BUD23 were different than in normal cells, and that cells without BUD23 produced different proteins, which did not always perform their roles correctly. Proteins in the mitochondria are one of the main groups affected by the absence of BUD23. To determine what effects these modified mitochondrial proteins would have in an animal, Baxter et al. genetically modified mice so that they no longer produced BUD23. These mice developed heart problems caused by their mitochondria not working correctly and being unable to provide the energy the heart cells needed, eventually leading to heart failure. Heart problems are common in people with Williams-Beuren syndrome.

Many diseases arise when a person's mitochondria do not work properly, but it is often unclear why. These experiments suggest that low levels of BUD23 or faulty ribosomes may be causing mitochondria to work poorly in some of these diseases, which could lead to the development of new therapies.

_2009_). Protein synthesis is the most energetically-demanding process for a cell to perform and, therefore, mRNA translation is closely coupled to mitochondrial function (_Morita et al., 2013_).

Mitochondria are fundamental cellular components of eukaryotes, generating the majority of cellular ATP. The human mitochondrial genome contains only 37 genes, of which 13 are subunits of respiratory complexes, 22 encode mitochondrial tRNAs, and a further two encode rRNA. Whilst the mitochondrion translates the 13 protein-coding genes of its own genome using a bespoke mitochondrial apparatus, the majority of the genes required for a functional mitochondrion are encoded in the cell nuclear genome and are, therefore, dependent on the cytosolic translational apparatus. This relationship implies a fundamental role for the eukaryotic 80S ribosome in control of mitochondrial abundance and function. Indeed, mitochondrial content and function varies between tissues and cell-types, as well as within a tissue, in response to external and internal stimuli including energy demand, oxidative stress and cellular signals (_Palmer et al., 2011_). For example, red blood cells contain no mitochondria whereas in cardiomyocytes mitochondria occupy 30% of cell volume to meet the exceptionally high ATP demand (_Piquereau et al., 2013_). Mitochondrial dysfunction has been linked to a wide range of cardiac disorders, often due to the aberrant production of ROS, and attendant cell death (_Kanaan and Harper, 2017_; _Ott et al., 2007_).

BUD23 (previously known as WBSCR22) was initially identified as a putative methyltransferase implicated in tumour metastases (_Nakazawa et al., 2011_). Its expression is responsive to diverse inflammatory and cancer pathologies (_Jangani et al., 2014_), and reduction of BUD23 expression can affect cellular response to glucocorticoid and alter histone methylation (_Jangani et al., 2014_). However, BUD23 actions were diverse, and a unifying mechanism of action was elusive. More recent studies identified ribosomal RNA as the preferred substrate of BUD23 (_Haag et al., 2015_; _White et al., 2008_; _Zorbas et al., 2015_). There is little understanding of the physiological role of BUD23 in a mammalian context although it is one of the genes deleted in Williams-Beuren syndrome (WBS). WBS patients have a complex phenotype with prominent neurological and morphological features. In addition metabolic pathologies exist including diabetes and obesity (_Morris, 1993_). The contribution of individual genes within the 22-gene interval to the human phenotype has not been determined, but the neurological and metabolic features suggest a bioenergetic contribution.

BUD23 is a ribosomal RNA methyltransferase which imparts a methyl mark on a key guanosine residue (forming N7-methylguanosine) located between the E and P site of the small ribosomal sub-unit that has been mapped to residue G1575 of yeast 18S rRNA and G1639 of human 18S rRNA (*Haag et al., 2015*; *Õunap et al., 2013*; *White et al., 2008*; *Zorbas et al., 2015*). BUD23 protein is found in both the nucleus and the cytoplasm (*Õunap et al., 2015*), and its depletion leads to a nuclear accumulation of 18SE-pre-RNA (*Haag et al., 2015*). In both yeast and human cells the meth-yltransferase catalytic activity of BUD23 is not required for the processing of 18S pre-RNA, or the synthesis of 40S subunits, indicating that BUD23 has an additional role, distinct from its methyltrans-ferase activity (*Haag et al., 2015*; *Zorbas et al., 2015*).

In human cell lines it has been suggested that some 18S rRNA lacks the m7-G1639 mark, which may indicate a selective role in ribosome function (*Haag et al., 2015*). BUD23 stability requires inter-action with TRMT112 (TRM112 in yeast), an obligate binding partner, which is known to stabilise four client methyltransferase enzymes, all involved in the generation of the translational apparatus (*Bourgeois et al., 2017*; *Létoquart et al., 2014*). The binding of TRMT112 with these four methyl-transferases is competitive, which results in tight regulation of BUD23 protein concentration and enzymatic function at the level of protein stability.

Here we identify a novel mechanism linking the energy-demanding process of protein translation to mitochondrial dynamics through BUD23. We examine the role of BUD23 in ribosome function and discover that BUD23 preferentially promotes the selection of mRNA species with low GC-content 5'UTRs. We also identify a role for BUD23 in mitochondrial transcript translation which impacts mito-chondrial function in vitro. We go on to examine the role of BUD23 in a murine in vivo system and discover that the production of mitochondrial proteins is dependent on BUD23. Finally, we examine the role of BUD23 in the mitochondrially-rich and energetically-demanding cardiac tissue and dis-cover a cardiomyopathy phenotype leading to premature death. These discoveries identify BUD23 as essential for mammalian mitochondrial function, with implications for human mitochondrial dis-ease and cardiomyopathy.

## Results

### BUD23 plays a major role in translational homeostasis

We previously identified pleiotropic actions of BUD23 in airway epithelial cells, and attributed these to an epigenetic effect (*Jangani et al., 2014*). The discovery that BUD23 modifies ribosomal RNA rather than histone proteins prompted re-examination of the BUD23 role in physiology. To identify candidate pathways affected by BUD23 we depleted expression in human airway epithelial cells (*Figure 1A,B*). This intervention caused a small but statistically significant decrease in cell prolifera-tion (*Figure 1C*), which is similar to the anti-proliferative effects caused by BUD23 deletion in yeast (*White et al., 2008*).

To investigate comprehensively the role of BUD23 in ribosome function we performed polysome profiling. The 48 hr knockdown of *BUD23* resulted in a reduction of the 40S subunit peak, and a con-comitant increase in the 60/80S peak (*Figure 1D,E*). There was, however, little change in the profile of the polysome fractions at this early time-point following transient *BUD23* knockdown, allowing us to investigate the changes in mRNA substrate selection in the intact polysomes. To test the global efficiency of the translational apparatus in cells lacking BUD23, we used 35-S-methionine incorpo-ration for 1 hr (*Figure 1—figure supplement 1*). This demonstrated little overall impact on global protein translation rate, confirming that the identified polysomes were essentially competent at this time-point, despite the impact on 40S maturation. In order to determine if BUD23-loss differentially impacted the translation of a specific subset of mRNAs, we profiled the translational efficiency (TE) of individual mRNA species within the polysome profiles.

To do this, the polysome fractions were divided into 'heavy' (more than three ribosomes) or 'light' (less than three ribosomes) and pooled, prior to RNA extraction and sequencing. This defined a total of 14,527 transcripts, from which the relative proportion of each individual transcript in the heavy compared to the light fraction was calculated to give a translation efficiency (TE) score. Average TE scores across the three replicates for each condition (+ /- BUD23) were then plotted against each other (*Figure 1F*). This revealed that the most efficiently translated transcripts in the control samples (more heavily loaded with ribosomes), were most significantly reduced in the *BUD23* knockdown

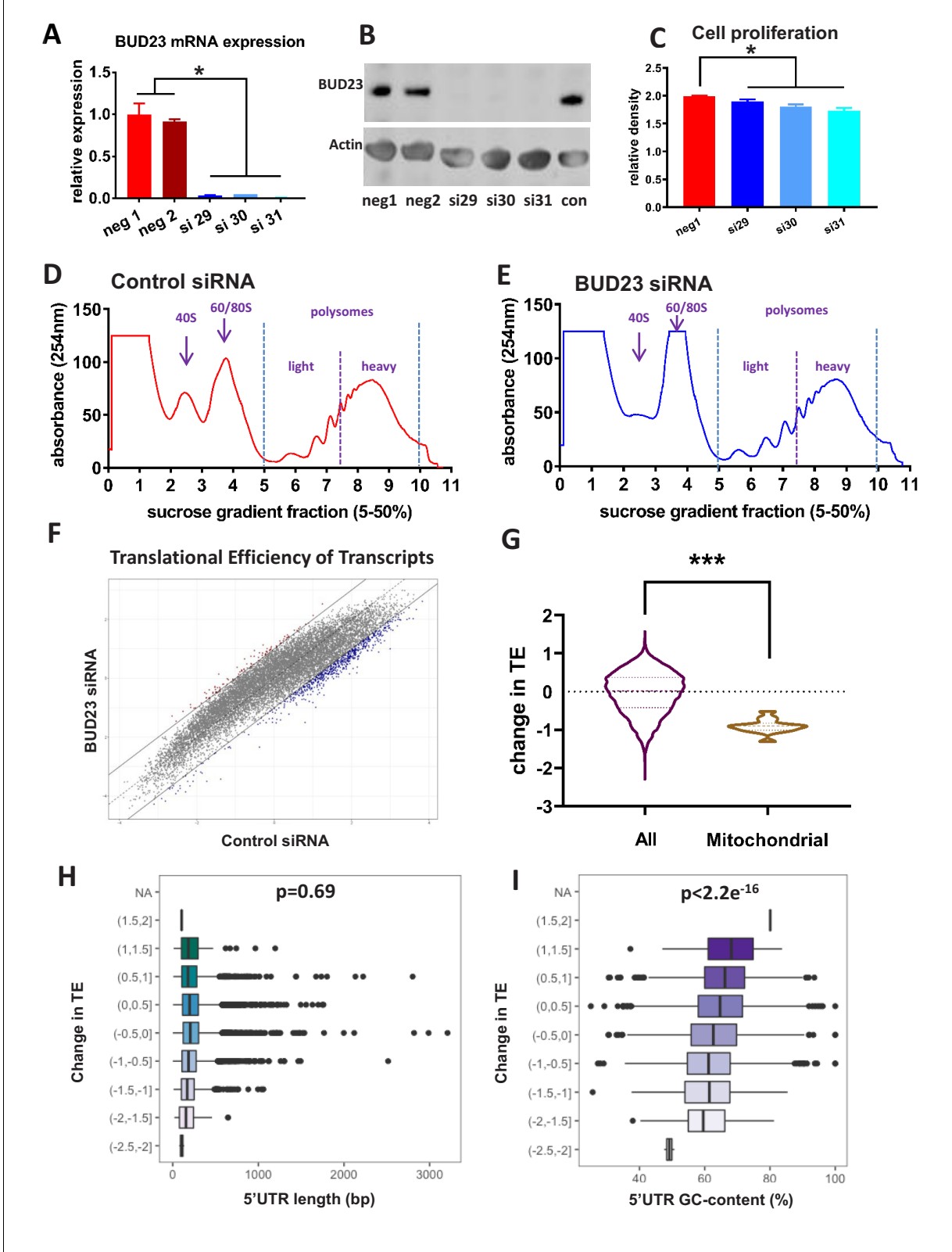

**Figure 1.** Analysis of BUD23-dependent ribosome function. (**A**) Relative mRNA expression levels of *BUD23*. A549 cells were transfected for 48 hr with either *BUD23* specific siRNA (si29, 30, 31) or non-targeting control (neg1, 2). Total RNA was extracted before reverse transcription to cDNA and qPCR (TaqMan). (**B**) Protein expression was determined by western blotting using antibody GTX105840. Actin was used as a loading control. (**C**) siRNA treated cells were plated into cell culture wells and allowed to grow in DMEM supplemented with 10% FBS for 48 hr. Total cell volume was then

*Figure 1 continued on next page*

*Figure 1 continued*

measured using SRB assay to indicate amount of proliferation. Polysome profiles were obtained from A549 cell lysates treated with (D) Control siRNA or (E) *BUD23* specific siRNA (si30) for 48 hr (n = 3). Changes in the ratio of the 40S peak to the 60/80S peak were observed between the two conditions. Polysome fractions were divided into heavy (more than three ribosomes) or light (three or less ribosomes) before RNA sequencing.. A translational efficiency (TE) score was derived by dividing the relative proportion of transcript abundance in the heavy fraction by that in the light fraction. (F) The TE scores in the two conditions derived from the samples in A and B are plotted against each other. (G) Violin plot showing the change in translational efficiency upon *BUD23* siRNA treatment ('change in TE score') for all detected transcripts (All) versus mitochondrially-encoded transcripts (Mitochondrial). (H) 5'UTR length of the transcripts was plotted against change in translational efficiency (TE, as defined in *Figure 1G*). The data are plotted in bins with 'change in TE' (BUD23 TE/Control TE) windows of 0.5. (I) 5'UTR GC content was of the same transcripts was plotted against change in TE. The data are plotted in bins with TE windows of 0.5.

The online version of this article includes the following figure supplement(s) for figure 1:

**Figure supplement 1.** Further analysis of BUD23-dependent ribosome function in A549 cells.
**Figure supplement 2.** 5'UTR motif analysis.
**Figure supplement 3.** Further analysis of polysome profile dataset.

condition. To examine the identity of the most highly affected transcripts, a threshold of >1 or <-1 difference in the relative TE was set (i.e. TE doubled or halved), and applied to *Figure 1F*, shown as trend lines. There were 650 transcripts with a TE ratio <-1 and 95 transcripts where the TE ratio >1 after BUD23 depletion; indicating the marked loss of high efficiency mRNA translation.

## Analysis of BUD23 differentially translated mRNA species

The 650 transcripts for which TE was reduced (TE ratio <-1 after BUD23 depletion) constituted a highly connected network, with a significantly higher number of protein-protein interactions than would be expected from a random dataset of similar size, and an interaction analysis (PPI) enrichment p-value<1e$^{-16}$. Gene set analysis of these transcripts revealed enrichment for genes involved in RNA processing, metabolic processes and organelle organisation among other high energy demand processes (*Figure 1—figure supplement 1*).

Surprisingly, the TE of mitochondrially-encoded transcripts was observed to be particularly strongly down-regulated (*Figure 1G*). In fact, all mitochondrial transcripts had a negative change in TE after BUD23 loss, with approximately 50% reduction in TE.

## BUD23 expression promotes ribosome interaction with low GC 5'UTRs

To investigate the mechanism underlying the BUD23 dependent selection of transcripts we examined specific mRNA features associated with translational efficiency. Given the known influence of 5'UTR features on translation efficiency, we reasoned that specific features of the 5'UTR may be particularly important for BUD23 action. A number of well-characterised 5'UTR features impact mRNA translation, including length, secondary structure, GC content, and specific motifs including the TOP sequence (*Gandin et al., 2016*; *Meijer and Thomas, 2002*). We saw no enrichment for TOP sequence content in the differentially translated mRNA species (*Figure 1—figure supplement 2*), and no correlation between change in TE score and 5'UTR length was observed (*Figure 1H*). We also noted that transcripts with shorter total lengths were not more likely to shift from the heavy to the light fraction (*Figure 1—figure supplement 3*). There was, however, a highly significant correlation between change in TE score and GC content (Pearson's correlation value of −0.1337, p-value<2.2e$^{-16}$) (*Figure 1I*). This identifies an intersection between mRNA transcript 5'UTR GC content and BUD23 ribosome maturation. There was no difference in the distribution of 5'UTR GC in mitochondrial-related transcripts relative to all transcripts detected in the polysome analysis (*Figure 1—figure supplement 3*).

## Loss of BUD23 causes changes in cellular protein homeostasis

Our studies reveal the consequences for the ribosome of BUD23 action and point to an impact on the cellular proteome. Therefore, to investigate the downstream consequences of BUD23 loss we examined the cellular proteome by LC-MS/MS. *BUD23* knockdown downregulated 83 proteins and upregulated 64, out of a total of 3255 identified (*Figure 2—source data 1*). When tested using PANTHER GO cellular component ontology analysis, upregulated proteins were over-represented for components of the large ribosomal subunit (14/64 proteins), whilst downregulated proteins were

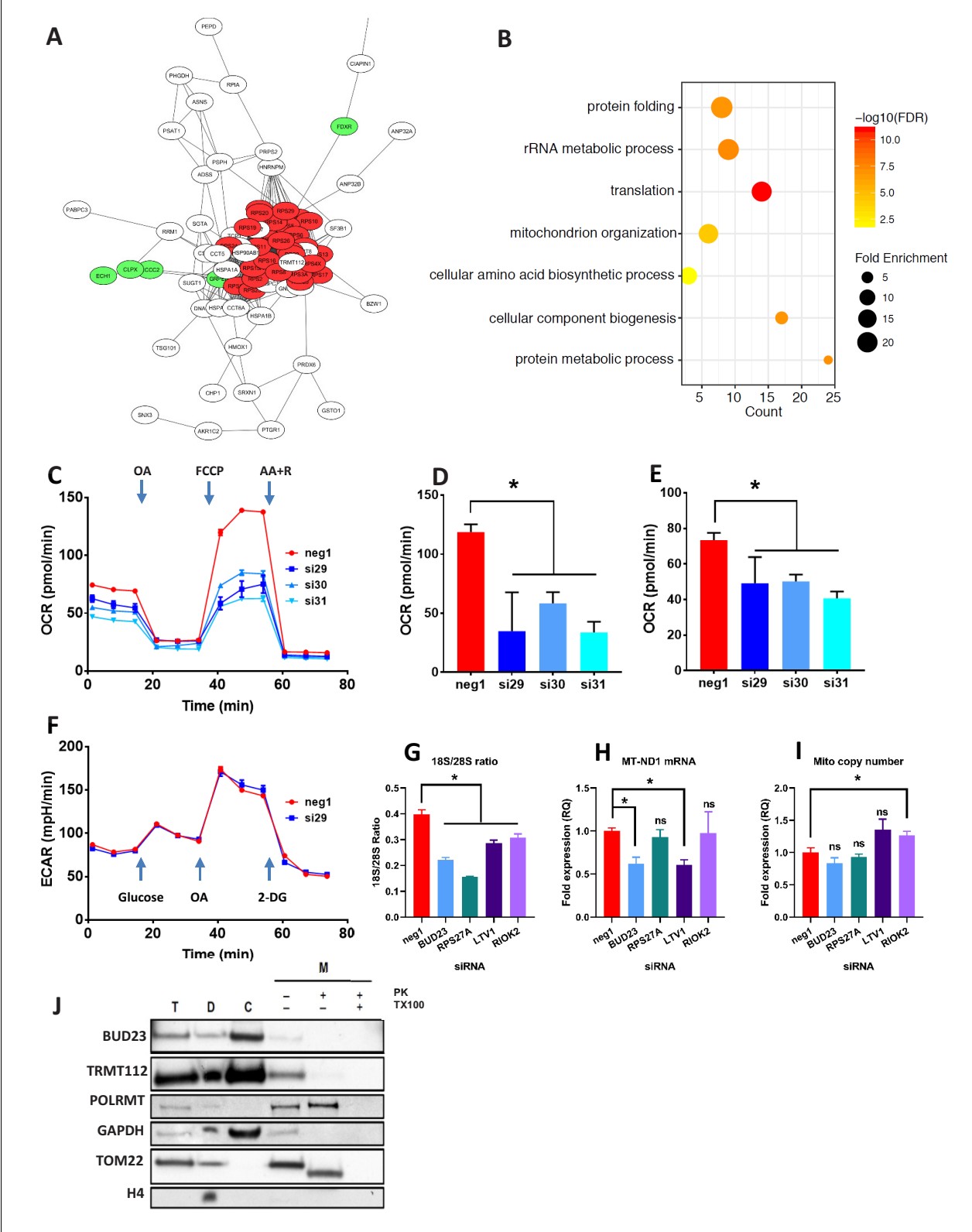

**Figure 2.** Proteomic analysis of BUD23 deficiency reveals mitochondrial phenotype Proteomics was performed on A549 cell lysates 48 hr after transfection with either *BUD23*-targeting siRNA or control siRNA. Significantly down-regulated and up-regulated proteins were defined using Welch's T-test with a permutation-based FDR < 0.05. (**A**) Significantly down-regulated proteins were visualised as a network using String 10 online software. Red nodes are 40S ribosomal proteins, Green nodes are mitochondrial proteins (**B**) Over-representation analysis of the significantly down-regulated protein

*Figure 2 continued on next page*

*Figure 2 continued*

dataset was performed using the PANTHER GO-Slim Biological Processes, and significantly over-represented terms were visualise using ggplot2. (**C**) Mitochondrial stress test was conducted on A549 cells treated with *BUD23*-targeting or non-targeting control siRNA (n = 3). Oxygen consumption rate (OCR) was measured over time to obtain mitochondrial respiration rates. Values for ETS (**D**) and OXPHOS (**E**) were derived from the OCR measurements after the addition of oligomycin A (OA), FCCP and antimycin A + rotenone (AA+R) in succession. (**F**) Glycolytic capacity was tested using a Seahorse glycolytic stress test and measuring extracellular acidification rate (ECAR). (**G**) A549 cells were treated with control siRNA (neg1) or siRNA specifically targeting *BUD23* (si30), RPS27A, LTV1 or RIOK2. 18S/28S ratio was measured using Tapestation to confirm ribosome subunit imbalance. (**H**) MT-ND1 mRNA was measured by qPCR as an indicator of mitochondrial transcript abundance (relative to control gene HPRT). (**I**) Mitochondrial copy number was measured relative to genomic copy number by qPCR. (**J**) Cellular fractionation into total 'T', cytosolic 'C', cellular debris 'D' and mitochondrial 'M' fractions. Protein was isolated from these fractions before SDS-PAGE and immuno-blotting for BUD23 and TRMT112. POLRMT was blotted as a control for mitochondrial proteins, TOM22 as a control for a mitochondrial membrane protein and GAPDH and Histone H4 as control non-mitochondrial proteins.

The online version of this article includes the following source data and figure supplement(s) for figure 2:

**Source data 1.** Proteomics data table.
**Figure supplement 1.** Network analysis of up-regulated proteins.
**Figure supplement 2.** Further analysis of ribosomal deficiency in A549 cells.

over-represented for components of the small ribosomal subunit (22/83 proteins), (*Figure 2A*, *Figure 2—figure supplement 1*). This imbalance in the relative abundance of ribosomal proteins, recapitulates the observation in the polysome profiles and confirms a major role for BUD23 in the maturation of the protein translation apparatus of the cell. Interestingly, TRMT112 protein, an obligate partner for BUD23, was also significantly down-regulated with *BUD23* knock-down, which may indicate a reciprocal stabilising interaction.

Ontology analysis of the proteomics dataset was used to predict the functional consequences of BUD23 reduction (PANTHER GO biological processes). Over-representation analysis of the up-regulated proteins overlapped significantly with the 'protein translation'. Analysis of the down-regulated proteins again showed a statistically significant over-representation for proteins associated with GO biological terms including 'translation', as well as 'rRNA metabolic process' and 'protein metabolic process' which support a major effect on the translational apparatus (*Figure 2B*). Unexpectedly, there was also over-representation within the down-regulated proteins for the terms 'Mitochondrion organization' and 'Cellular component biogenesis', suggesting a consequential impact on mitochondrial function, a major destination for new protein synthesis in the cell.

## BUD23 regulates mitochondrial function

Because the TE of mitochondrial transcripts was down-regulated and mitochondrial organisation emerged from the proteome analysis, specific assays of mitochondrial activity were performed. Depletion of BUD23 with any of three, independent siRNAs resulted in a > 50% reduction in oxidative phosphorylation (OXPHOS) and around a 25% reduction in Electron Transfer Capacity (ETC) (*Figure 2C,D,E*) but no observable difference in leak respiration (LEAK), relative to control (*Figure 2—figure supplement 2*). Moreover, no alteration in glycolytic capacity was observed indicating specificity for mitochondrial energetic pathways (*Figure 2F*). We also observed a reduction in mitochondrial mRNA abundance with *BUD23* knockdown, but no change in mitochondrial genome copy number, implying a functional defect, rather than loss of mitochondrial mass (*Figure 2G,H,I*), although these measurements were performed soon after siRNA knockdown, and so a later impact on mitochondrial mass resulting from prolonged BUD23 loss cannot be excluded.

To examine whether this mitochondrial transcription defect was specific to BUD23 deficiency, or part of a more general mechanism resulting from 40S/60S imbalance, we performed siRNA-mediated knockdowns of other known ribosome biogenesis factors, LTV1 and RIOK2, as well as the ribosomal small subunit protein RPS27A (also reported to result in ribosomal subunit imbalance) (*Sloan et al., 2019*). Depletion of *LTV1*, *RIOK2* and *RPS27A* all resulted in significant decrease in 18S/28S ratio relative to control siRNA, indicating a similar 40S/60S imbalance as observed with *BUD23* knockdown (*Figure 2G*, *Figure 2—figure supplement 2*). *LTV1* knockdown resulted in a similar decrease in mitochondrial transcript expression as observed in *BUD23* knockdown (*Figure 2H*). However, *RPS27A* and *RIOK2* knockdown-induced ribosome imbalance did not affect mitochondrial

transcript expression (*Figure 2H*, *Figure 2—figure supplement 2*). This indicates that disruption of 40S subunit biogenesis is not by itself sufficient to result in mitochondrial dysfunction.

To check that BUD23 was not regulating mitochondria directly, we blotted for the protein after sub-cellular fractionation. BUD23 was found to be abundant in the cytosolic fraction but was not detected in the mitochondrial fraction after proteinase K digestion (*Figure 2J*). This suggests the mechanism of BUD23 action does not occur within the mitochondria. Furthermore, TRMT112, the obligate BUD23 binding partner, was also abundant in the cytosol, but barely detectable within mitochondria (*Figure 2J*). Together these data indicate that BUD23-dependent mitochondrial regulation is likely to be a downstream consequence of effects on ribosome composition, function, and cellular protein repertoire, rather than a direct effect of BUD23 protein within the mitochondria.

## Homozygous deletion of *Bud23* results in embryo-lethality

To investigate the physiological impact of BUD23 we generated *Bud23* null mice. We introduced a frameshift deletion within the *Bud23* gene, which resulted in a null allele (*Figure 3A*). However, upon further breeding of heterozygous mice a clear deviation from the expected Mendelian ratio was observed in the resultant offspring. Out of 74 pups born, 39 were found to be wild-type for the *Bud23* gene, 35 were heterozygous, and zero were homozygous for the null allele (*Figure 3B*), indicating embryonic lethality in *Bud23*-null mice. This is in contrast to reports in yeast where BUD23 was shown to be non-essential for life (*White et al., 2008*). Furthermore, it is notable that approximately half the expected heterozygote birth rate was seen, implying an additional haplo-insufficiency developmental death rate. Accordingly, we analysed embryos at day E10.5 and detected Mendelian ratios of wild-type, and heterozygous animals, but again no homozygous null embryos were seen (*Figure 3B*), suggesting total *Bud23* loss is incompatible with embryogenesis, but haploinsufficiency results in fetal death later in development.

Surprisingly, surviving adult heterozygous mice showed no difference compared to wildtype littermate controls in body weight, lean body weight or body fat between 10 and 30 weeks of age (*Figure 3C*). There was also no significant change in energy expenditure or respiratory exchange ratio in *Bud23*[+/-] mice relative to WT littermate controls (*Figure 3—figure supplement 1*), suggesting compensatory mechanisms to permit survival.

To circumvent embryonic lethality, we generated a floxed *Bud23* allele mouse, allowing post-natal tissue-specific KO (*Figure 3A*). Mice homozygous for this floxed *Bud23* (*Bud23*[fl/fl]) allele showed no observable differences from wild-type littermates. To confirm the embryonic lethality phenotype using an independent genetic approach, we crossed these mice with a Cre-deleter mouse line (*Hprt*-cre) (*Nichol et al., 2011*) to globally delete *Bud23* alleles. *Bud23*[+/-] heterozygous crosses also failed to generate any homozygous null offspring (0 out of 28 cre positive offspring), confirming the observation in the global knockout mouse line (*Figure 3D*).

We targeted *Bud23* disruption to skeletal and cardiac muscle using Muscle Creatine Kinase (*Mck*)-cre as the driver (*Whitnall et al., 2008*). Cardiac muscle relies heavily on mitochondrial production of ATP in post-natal life, but less so during development, so offering us the chance to analyse BUD23 impact on a highly mitochondrially-dependent tissue. *Bud23*[fl/fl] *Mck*-Cre[+/-] mice exhibited Cre-mediated deletion of exon 7 of the *Bud23* gene in cardiac tissue, with a consequent reduction in BUD23 protein levels (*Figure 3E,F*).

## BUD23 deficiency drives preferential dysregulation of mitochondrial proteins

Loss of BUD23 in cardiac muscle resulted in sudden death between the age of 28–35 days (*Figure 4A*). Littermate control mice expressing *Bud23*[fl/fl]*Mck*-Cre[-/-] or *Bud23*[fl/wt] *Mck*-Cre[+/-] (i.e. mice retaining at least one functional copy of the *Bud23* allele) were viable, fertile and did not die prematurely (*Figure 4A*). Mouse tissues from this line were therefore subsequently harvested at 26 days of age. A significant reduction in the 18S/28S RNA ratio, consistent with the in vitro data using siRNA knockdown was observed at this time point (*Figure 4B*).

Proteomic analysis of cardiac tissue recovered from 26 day old *Bud23*[fl/fl]*Mck*-Cre[+/-] mice and littermate controls (*Bud23*[fl/fl]*Mck*-Cre[-/-]) revealed a significant decrease in protein content per cell with loss of BUD23 (*Figure 4C*), with clear separation by genotype (*Figure 4—figure supplement 1*). When detected proteins were annotated by sub-cellular compartment, mitochondria were found to

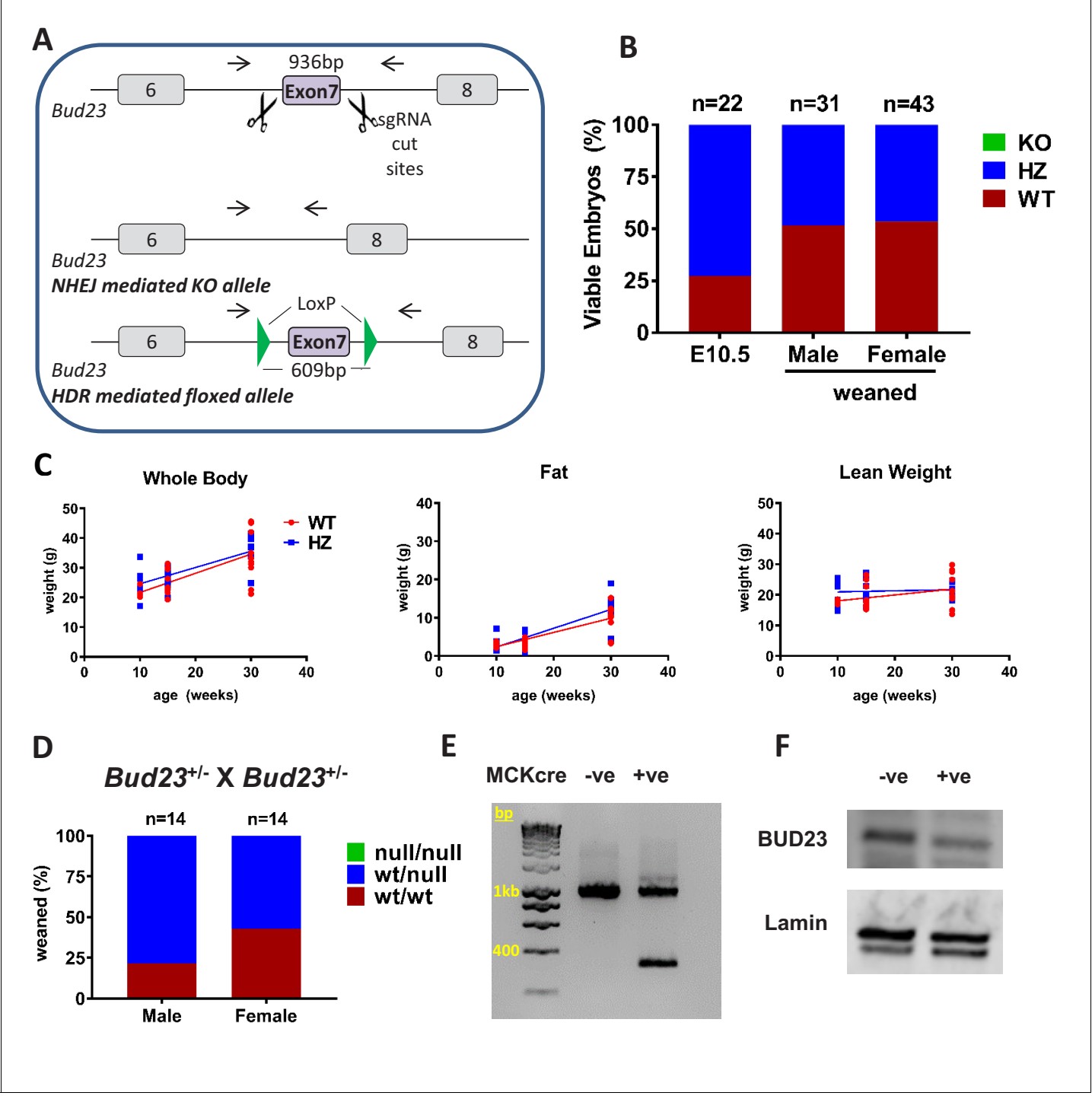

**Figure 3.** Generation and characterisation of murine *Bud23* null and floxed alleles. (**A**) Schematic of the targeting strategy used to create a murine *Bud23* knockout allele and a floxed allele were generated by CRISPR-Cas9. (**B**) Genotype analysis of day E10.5 embryos and the offspring from heterozygous breeding pairs. (**C**) *Bud23* Heterozygous (HZ) and wildtype (WT) mice (10–30 weeks old) were analysed over time for body composition by ECHO-MRI. (**D**) Genotype analysis of offspring from heterozygous recombined *Bud23* (*Bud23*rec/wt) allele crosses. (**E**) PCR amplification of the *Bud23* genomic locus in *Bud23*fl/fl mice positive or negative for MCKcre expression in heart tissue, shows successful recombination of the mutant *Bud23* allele (bp = base pairs). (**F**) Disruption of BUD23 protein in the cre positive animals was confirmed by Western blot.

The online version of this article includes the following figure supplement(s) for figure 3:

**Figure supplement 1.** Metabolic analysis of BUD23 heterozygous mice.

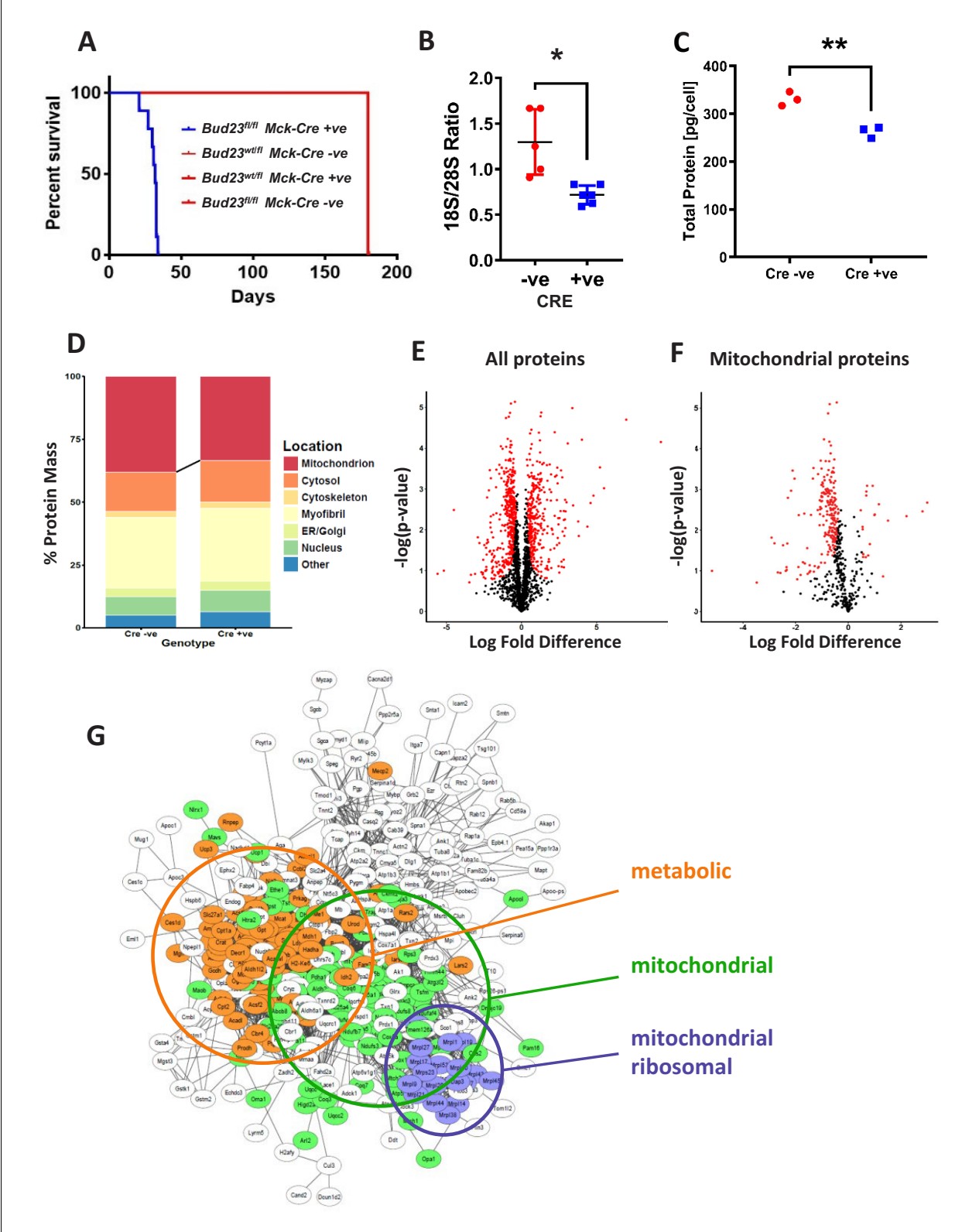

**Figure 4.** Proteomic analysis of BUD23 deficient heart tissue shows selective mitochondrial phenotype A murine *Bud23* knockout allele and a floxed allele were generated by CRISPR-Cas9. (**A**) Survival curve for *Bud23*fl/fl *Mck*-Cre+/− (*Bud23* knock-out targeted to heart and muscle) shown in blue, compared to littermate controls (*Bud23*wt/fl *Mck*-Cre−/−, *Bud23*wt/fl *Mck*-Cre+/−, *Bud23*fl/fl *Mck*-Cre−/−), all shown in red. For subsequent analyses, hearts were collected from 26 day old *Bud23*fl/fl *Mck*-Cre+/−, and *Bud23*fl/fl *Mck*-Cre−/− littermate controls. (**B**) 18S/28S ratio analysis of RNA extracted from

*Figure 4 continued*

BUD23 positive or negative mouse hearts using TapeStation. (**C**) Proteomics was performed on heart tissue from Cre +ve and Cre -ve littermates (n = 3). Total protein concentration per cell is shown, as determined via the proteomic ruler approach. (**D**) Relative contribution levels of each organelle to total protein mass based on ontology. The decrease in mitochondrial mass observed in Cre positive hearts is indicated. (**E and F**) Volcano plots of either total protein or mitochondrial protein fold change differences in Cre positive hearts relative to Cre negative hearts. Significantly up- and down-regulated proteins are indicated in red (FDR < 0.05, s0 = 0.2). (**G**) Functional enrichment network of total down-regulated proteins. The network was built using String 10 online software prior to export into Cytoscape.

The online version of this article includes the following source data and figure supplement(s) for figure 4:

**Source data 1.** Proteomics data table.
**Figure supplement 1.** Proteomic analysis of BUD23 deficient mouse hearts.
**Figure supplement 2.** Change in TE correlates poorly with change in protein abundance.
**Figure supplement 3.** Network analysis of down-regulated proteins in BUD23 deficient mouse hearts.

be subject to the largest loss of protein content, with small reciprocal increases in the other compartments (*Figure 4D*), again identifying mitochondria as especially sensitive to BUD23 action. Of the 2047 proteins identified by mass spectroscopy, 347 were found to be significantly up-regulated and 442 were found to be significantly down-regulated in the BUD23 deficient condition (*Figure 4E*, *Figure 4—source data 1*). Whilst an approximately equivalent number of down and up regulated proteins was identified globally, when this list was refined to mitochondrial proteins only 20 were up-regulated compared to 220 down-regulated (50% of all down-regulated proteins) (*Figure 4F*). By contrast, the up-regulated dataset was enriched for translational (60S proteins) and proteasomal proteins (*Figure 4—figure supplement 1*). Whilst correlation between change in protein abundance in BUD23-deficient heart tissue and change in mRNA TE of the corresponding transcripts in the *BUD23* siRNA treated A549 cells was poor overall (*Figure 4—figure supplement 2*), it is striking that mitochondria are affected to a great extent in both models. These observations show a strong selectivity for impaired expression of mitochondrial proteins as a result of BUD23 deficiency.

Within the set of mitochondrial proteins, we found that electron transport chain complexes I, IV and V (ATP synthase) were most strongly reduced in response to loss of BUD23 (*Figure 4—figure supplement 3*). Furthermore, ontology analysis of all the significantly down-regulated proteins identified three main clusters of proteins: mitochondrial proteins, mitochondrial ribosomal proteins and proteins involved in energy metabolism, indicating a functional mechanism linking between the BUD23 dependent ribosomal defect, through reduced mitochondrial protein expression to result in disruption of energy metabolic pathways (*Figure 4G*). Therefore, BUD23 actions on the ribosome result in down-regulation of core mitochondrial proteins involved in ATP synthesis.

## BUD23 deficiency causes mitochondrial dysfunction in cardiomyocytes

Given the observation of preferential dysregulation of mitochondrial proteins, we hypothesised that mitochondrial function was likely to be compromised in BUD23-deficient cardiac tissue. Mitochondrial DNA copy number was reduced in BUD23 deficient cardiac tissue (*Figure 5A*). Functional analysis of mitochondrial function in cardiac homogenates using the Oroboros microrespirometer system revealed that mitochondrial OXPHOS and ETC were both significantly reduced in mice deficient for BUD23, while LEAK respiration was unaffected (*Figure 5B*). The reduction in OXPHOS led to a lower respiratory control ratio (RCR) indicating that mitochondria from mice deficient for BUD23 were less efficient at producing ATP. These effects were apparent regardless of which substrate was used for electron donation. In addition to reduced mitochondrial capacity, mice deficient for BUD23 had lower citrate synthase activity, indicating a reduction in cardiac mitochondrial density (*Figure 5C*), which accords with the measured loss of mitochondrial genome copy number (*Figure 5A*).

To test whether haplo-insufficiency of BUD23 protein was enough to impair mitochondrial function we also tested heart homogenates from *Bud23*$^{+/-}$ mice. These were found to have no significant difference in mitochondrial respiratory capacity across any mitochondrial state (LEAK, OXPHOS, ETS), whether normalised to citrate synthase activity or protein content (*Figure 5—figure supplement 1*). It was, however, noted that, compared to wild-types, *Bud23*$^{+/-}$ mice did exhibit significantly higher citrate synthase activities, indicating greater mitochondrial density. This may result from increased mitochondrial biogenesis to compensate for defective oxidative phosphorylation function and may partly explain the reduced viability in *Bud23*$^{+/-}$ mice. Interestingly, the impact of BUD23

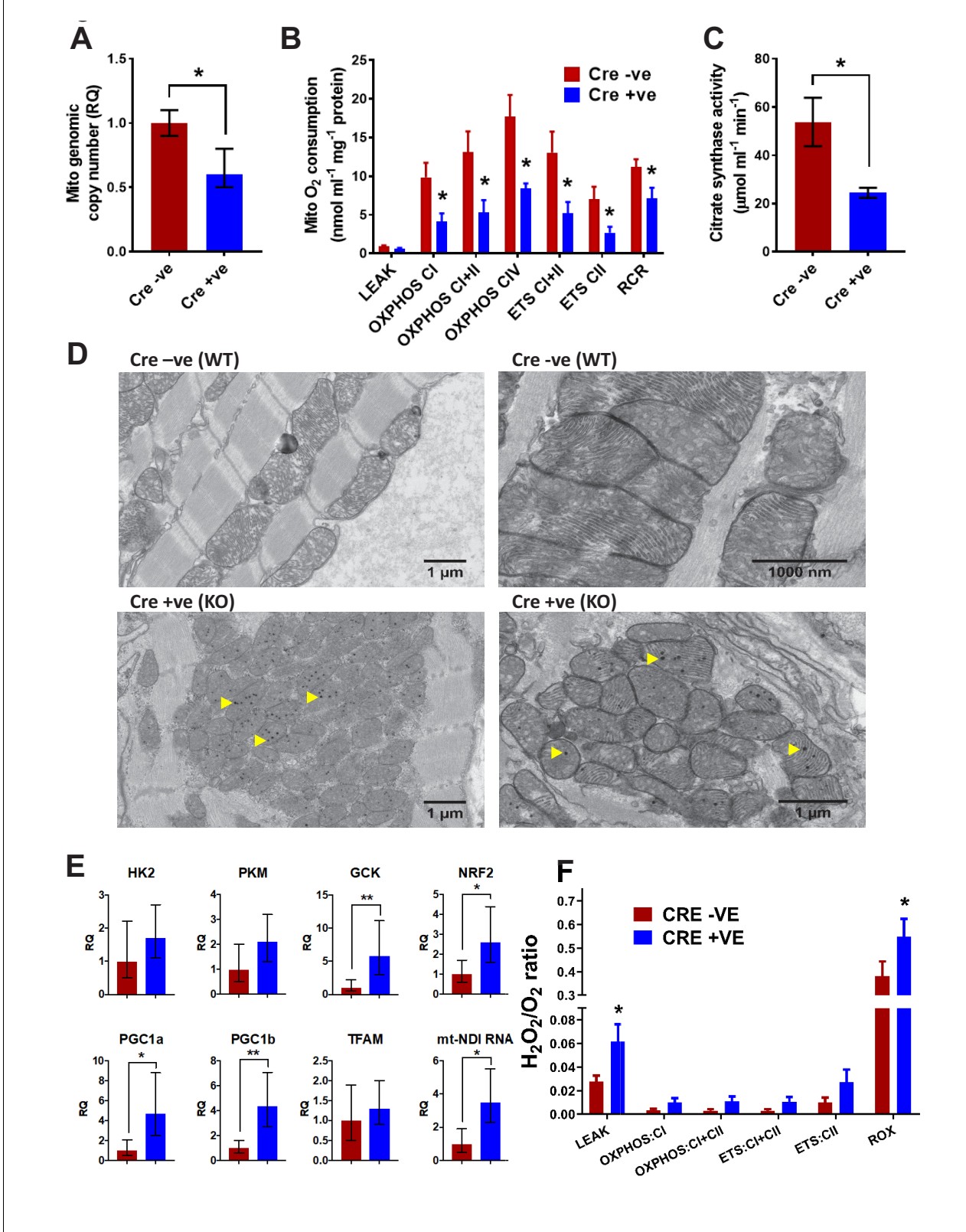

**Figure 5.** Analysis of mitochondrial structure and function in BUD23 depleted heart. Mouse tissues were collected at 26 days old. (A) Mitochondrial genome copy number measured in the heart tissue by qPCR, relative to autosomal genome copy number (n = 6). (B) Mitochondrial respiration was measured in heart homogenates (n = 5–6). Oxygen consumption was measured under various mitochondrial states and normalised to protein content: LEAK conditions (malate, pyruvate and glutamate as substrates); OXPHOS (routine respiration rate) through Complex I, I+II or IV; ETC (maximal

*Figure 5 continued*

respiration rate) through Complex I+II and II; Respiratory Control Ratio (RCR, measure of mitochondrial efficiency). (C) Citrate Synthase activity was measured in the heart homogenates as a marker of mitochondrial content. (D) Electron microscopy images of mitochondria in cardiomyocytes from either *Bud23*^fl/fl (WT) hearts, or *Bud23*^fl/fl *Mck*-Cre +ve (KO) hearts. Accumulation of electron dense granules is visible in *Bud23*^fl/fl *Mck*-Cre +ve (KO) hearts, denoted by arrows. (E) These hearts were further analysed for expression of mitochondrial biogenesis genes, markers of glycolysis, mitochondrial biogenesis, oxidative stress and indicators of mitochondrial transcription using qPCR. (F) ROS was measured simultaneously with $O_2$ consumption (see *Figures 1–3*) under various mitochondrial states and normalised to citrate synthase activity: LEAK conditions (malate, pyruvate and glutamate as substrates); OXPHOS (routine respiration rate) through Complex I or I+II; ETS (maximal respiration rate) through Complex I+II and II, and in the presence of antimycin-A (residual oxygen consumption, ROX)(n = 5–6).

The online version of this article includes the following figure supplement(s) for figure 5:

**Figure supplement 1.** Oroboros analysis of mitochondrial function in cardiac homogenates from HZ mice and WT littermate controls.

loss on mitochondrial function appears to be sexually-dimorphic. When mitochondrial respiration was normalised to citrate synthase, male heterozygous (HZ) mice exhibited a deficiency in OXPHOS and ETS relative to wildtype, whereas female mice did not (*Figure 5—figure supplement 1*). This observation indicates a more severe phenotype in BUD23 deficient males, with further derangement of individual components of the electron transport chain.

## Analysis of mitochondrial morphology, and operation of compensatory circuits in targeted cardiac muscle

Electron microscopy (EM) analysis revealed that despite decreased mitochondrial protein abundance and reduced mitochondrial function in the BUD23 cardiac tissue, the individual mitochondria appear to be formed normally with an equivalent number of cristae compared to the wildtype. The inter-myofibrillar mitochondria were typically arranged in a highly ordered pattern in the wildtype cardiac tissue, however, there was marked disorganisation in the BUD23-deficient cardiac tissue (*Figure 5D*). In addition, we identified the presence of numerous spherical electron dense inclusion bodies within the mitochondria of BUD23 deficient cardiac cells (*Figure 5D*, yellow arrows). Similar electron dense structures have been previously reported in myocardial mitochondria, which may accumulate in response to increased cellular bioenergetic demand (*Jacob et al., 1994*). The observation of these structures here may be consistent with attempted mitochondrial adaptation to energetic-demand challenge.

A frequent cellular adaptation to impaired oxidative phosphorylation is a switch to glycolytic ATP generation, and so we measured rate-limiting glycolytic gene expression (*Figure 5E*). We identified a significant induction of glucokinase (GCK), and a trend to induction of phosphofructokinase (PKM) and hexokinase (HK2) supporting such an adaptive switch to glycolysis. A further adaptation predicted was an induction in mitochondrial biogenesis programmes. We identified a significant induction in PGC1a and PGC1b, but no change in TFAM1 expression (*Figure 5E*). Taken together these results point to some of the expected cellular responses to loss of oxidative phosphorylation, but it was surprising that the changes were so few. This implies that additional cellular injury may be proceeding, which acts to limit successful adaptation.

Impaired mitochondrial function may result in overproduction of reactive oxygen species, which can trigger the apoptotic cascade and lead to cell death. Such a progression may explain the distorted and thinned ventricular walls observed in the cardiac *Bud23*-null mice. We found induction of NRF2 (*Figure 5E*), a key sentinel gene controlling cellular responses to reactive oxygen species stress, which led us to measure reactive oxygen species directly, and found that these were significantly higher in hearts lacking BUD23 expression (*Figure 5F*), identifying a burden of increased reactive oxygen species in BUD23 deficient cardiomyocytes resulting from the action of deranged mitochondria.

## BUD23 deficiency in cardiac muscle leads to dilated cardiomyopathy

Deficient mitochondrial generation of ATP in cardiac tissue is a key cause of cardiomyopathy. Given the strong mitochondrial deficiencies observed in BUD23-deficient cardiac tissue, we predicted that the cardiomyopathy was the likely cause of sudden death. Histological examination of hearts from *Bud23*^fl/fl *Mck*-Cre^+/- (BUD23-deficient) mice showed significant enlargement, ventricle dilatation and decreased ventricle wall thickness compared to littermate controls at 26 days old (*Figure 6A–D*).

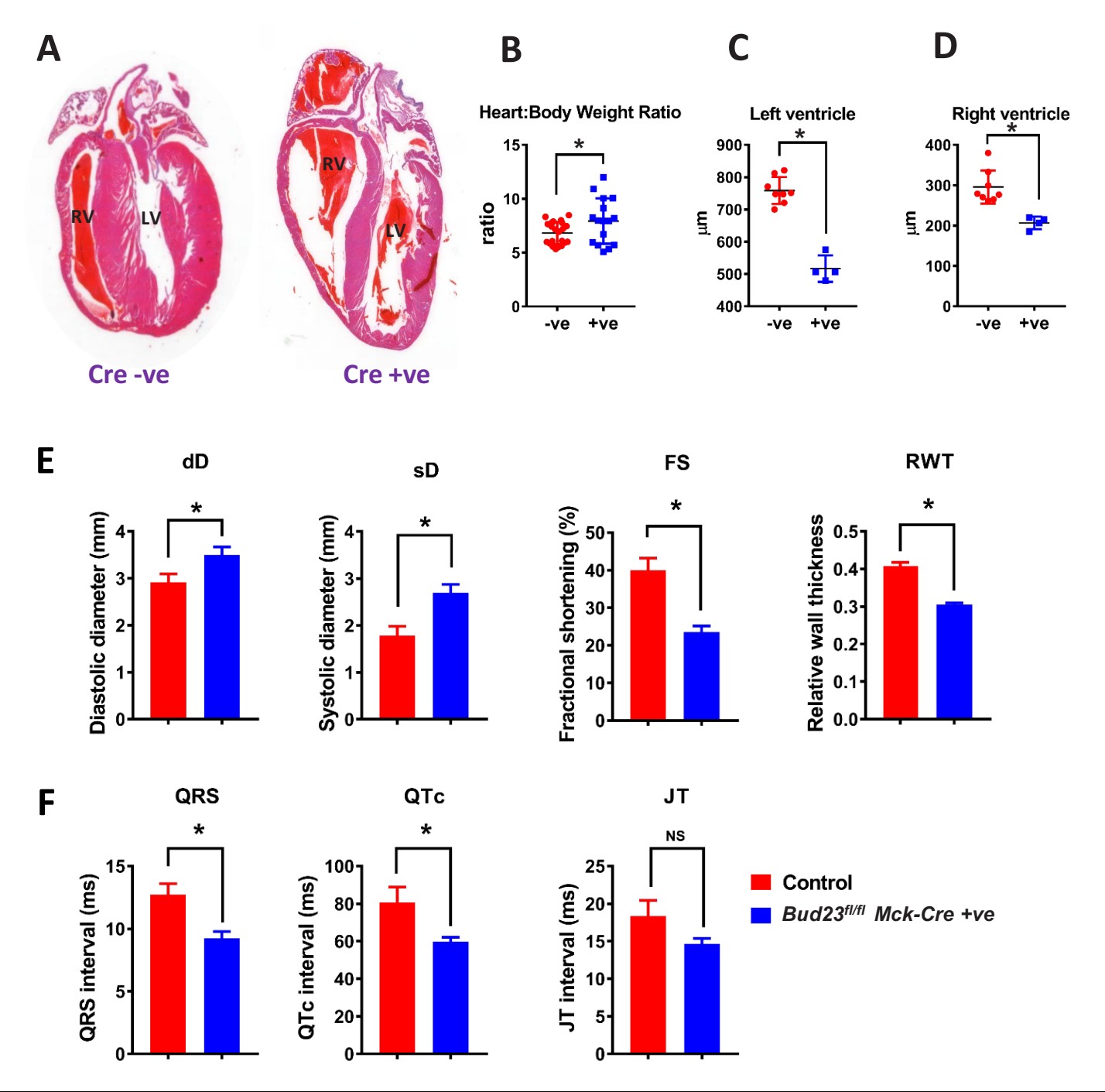

**Figure 6.** Targeted disruption of *Bud23* in cardiac tissue results in dilated cardiomyopathy. (**A**) Heart sections cut in the coronal plane and stained with haematoxylin and eosin from day 26 mice. Representative images are shown. (**B**) Heart weight measured as a ratio to total bodyweight. (**C and D**) Analysis of heart ventricle wall thickness in H and E stained heart sections. (**E**) Echocardiography of *Bud23*[fl/fl]*Mck*-Cre[+/-] mice and littermate controls at 26–28 days of age (n = 9–10). Mice were lightly anesthetized maintaining the heart rate at approximately 450 beats/min. Data for diastolic diameter (dD), systolic diameter (sD), fractional shortening (FS) and relative wall thickness (RWT) are shown; full parameter measurements are shown in supplementary. (**F**) Electrocardiography of *Bud23*[fl/fl]*Mck*-Cre[+/-] mice and littermate controls at 26–28 days of age (n = 9–10). Data for QRS, QTc and JT intervals are shown. Full parameter measurements are shown in supplementary. * denotes statistical significance (p<0.05) using Student's T-test.
The online version of this article includes the following source data and figure supplement(s) for figure 6:

**Source data 1.** Data table of echocardiogram analysis results and electrocardiogram analysis results.
**Figure supplement 1.** Fibrosis markers are up-regulated in BUD23 deficient mouse hearts.

Littermate control mice expressing $Bud23^{fl/fl}Mck\text{-}Cre^{-/-}$ or $Bud23^{fl/wt}$ $Mck\text{-}Cre^{+/-}$ (i.e. mice retaining at least one functional copy of the $Bud23$ allele) were viable, fertile and did not die prematurely (*Figure 3A*).

Echocardiogram analysis of $Bud23^{fl/fl}$ $Mck\text{-}Cre^{+/-}$ mice at 26–28 days old showed that cardiac BUD23-deficiency resulted in systolic dysfunction indicated by significantly dilated left ventricle (sD), as well as significantly decreased fractional shortening and relative wall thickness (*Figure 6E*, *Figure 6—source data 1*).). There was, however, no observed pulmonary oedema or liver oedema (*Figure 6—figure supplement 1*,). Further analysis of the heart proteomics data from *Figure 4* showed an increase in a number of markers of fibrosis, including Col4a2, CTGF, Galectin-3 and others (*Figure 6—figure supplement 1*). Taken together, these observations indicate that $Bud23^{fl/fl}$ $Mck\text{-}Cre^{+/-}$ mice exhibit the early stages of heart failure around 26–28 days of age. Electrocardiogram analysis of the same mice at the same age revealed significantly shorter QRS and corrected QT (QTc) intervals but no change in heart rate (*Figure 6F*; *Figure 6—source data 1*).).

## Discussion

BUD23 is tightly conserved through evolution and has recently emerged as a ribosomal RNA methyltransferase, although its physiological role in mammals is unexplored. It is one of the genes deleted in Williams-Beuren Syndrome, a complex multi-gene deletion syndrome with neurological, and energy homeostatic features, although the mediating role of individual genes to the phenotype has not been determined (*Morris, 1993*). We now identify BUD23 as playing a major role in regulating the translational efficiency of specific transcripts through its role in ribosome maturation. Indeed, translation of low 5'UTR GC-content mRNA species were particularly dependent on BUD23 expression, in contrast to the picture seen with simple ribosome deficiency in which transcripts with short/unstructured 5'UTRs were seen to be most affected (*Khajuria et al., 2018*). Furthermore, we identified a surprising role for BUD23 in maintaining mitochondrial oxidative phosphorylation capacity. As protein translation is tightly coupled to ATP demand this raised the possibility that BUD23-dependent ribosomal maturation is critical to link the two processes in living cells and tissues (*Morita et al., 2013*). Indeed, BUD23 loss greatly impaired mitochondrial ATP generation, both in vitro and in vivo. In mice, cardiomyocyte loss of BUD23 greatly impaired mitochondrial ATP generation leading to dilated cardiomyopathy and premature death, and global loss of BUD23 resulted in embryo-lethality.

BUD23 is thought to perform two functions in the generation of the translational apparatus: firstly, in the processing of pre-18S RNA into its mature form, and secondly imparting a m7-G methyl mark on a key residue. As the catalytic activity of BUD23 is not required for efficient pre-18S RNA processing, these two functions appear to be independent of each other (*Haag et al., 2015*; *Zorbas et al., 2015*). It is notable that both the BUD23 imparted m7-G methyl mark and structurally complex 5'UTRs are characteristic features of translation specific to Eukaryotes. We speculate the co-evolution of this regulatory 18S methyl mark with the need for more complex regulation of differential translation. Further work should focus on delineating the effect of BUD23-dependent 18S maturation, from the role of the methylation mark.

We used proteomics to examine the relative changes in protein abundance which result from targeted loss of BUD23 in cardiomyocytes. This revealed a selective loss of mitochondrial proteins, explaining the observed reduction in mitochondrial capacity. As BUD23 appears to have no direct role within mitochondria we propose that the mitochondrial phenotype results from impaired translation of nuclear-encoded genes with a role in mitochondrial homeostasis. Indeed, we found that the translational efficiency of a number of mitochondria-targeted as well as mitochondrial regulating proteins were BUD23-dependent. There are many potential candidates linking the change of translational efficiency to the mitochondrial phenotype. Mammalian target of rapamycin (mTOR), for example, is known to regulate mitochondrial activity and biogenesis by regulation of translation (*Morita et al., 2013*) and, interestingly, BUD23 loss resulted in impaired translation of mTOR itself. We also observed a marked decrease in the translational efficiency of mitochondrially-encoded transcripts.

We initially generated and characterised a frame shift-generating null allele mouse. The homozygote null animals were all lost prior to embryonic day e10.5 and we were unable to identify a cause but considered that a major defect in a mitochondrially–dependent organ such as heart may be a

plausible mechanism. We moved, therefore, to a conditional allele mouse, using which we confirmed the developmental lethality of the null allele by crossing with a global deleter-cre strain (Hprt-cre). Using the conditional allele mice with a muscle Cre driver we were able to profile the impact of BUD23 in a physiological role. Loss of BUD23 in heart and skeletal muscle resulted in early death from cardiac failure. This striking phenotype was accompanied by morphological changes including cardiac dilatation, ventricular wall thinning, and impaired ventricular contractility. There was evidence for an attempted switch to glycolytic ATP generation in the hearts, with induction of rate-limiting glycolytic enzyme genes; and also adaptation to conditions of oxidative stress with induction of NRF2. Mitochondrial dysfunction frequently imposes a burden of oxidative stress on affected cells, which can result in apoptosis, thereby explaining the loss of cardiac muscle seen. Detailed ultrastructure analysis of cardiac mitochondria revealed no major loss of mitochondrial volume, but there was a striking additional feature seen within the mitochondria: electron-dense, spheroid inclusions. These were not observed in control hearts, and are not typically seen in mitochondria from individuals with mitochondrial myopathies, but have been reported before under conditions of extreme ATP demand, and to result from bilayer budding from the inner cristae, to generate additional ATP generating surface area (*Jacob et al., 1994*; *Somlyo et al., 1974*). Therefore, we propose this may be a further adaptation to the impaired oxidative phosphorylation function within the mitochondria.

There is a growing recognition that transcriptional control mechanisms are insufficient to explain differentiated cell function. Recent advances have identified differential translation of mRNA species dependent on 5'UTR features as an important control mechanism, potentially coupling global changes in protein translation with appropriate upregulation of mitochondrial ATP generation to meet demand [2]. BUD23, a highly conserved ribosomal RNA methyltransferase which lies within a multigene interval, deleted as a cause of Williams-Beuren syndrome, plays a specific role in highly ATP-hungry cells, promoting translation of mitochondrial proteins, and impacting on oxidative phosphorylation. This function likely contributes to aspects of the human phenotype, in particular neurological dysfunction, glucose intolerance and obesity. Complete global loss of *Bud23* is embryonic lethal, and early post-natal death follows deletion of *Bud23* in the heart. The cardiac phenotype is severe, with marked mitochondrial dysfunction, and evidence of compensatory changes towards glycolytic ATP generation, and adaptation to oxidative stress. Therefore, BUD23 plays a critical role in coupling protein translation to mitochondrial function, with implications for mitochondrial diseases, and cardiomyopathies.

# Materials and methods

**Key resources table**

| Reagent type (species) or resource | Designation | Source or reference | Identifiers | Additional information |
|---|---|---|---|---|
| Gene (*Homo sapiens*) | BUD23 | | Gene ID: 114049 | Aliases: WBSCR22, MERM1 |
| Genetic reagent (*Mus musculus*) | *Bud23*null | This paper | *Bud23*null | Mutant allele: deleted exon 7 of BUD23 gene |
| Genetic reagent (*Mus musculus*) | *BUD23*fl/fl | This paper | *BUD23*fl/fl | Mutant allele: floxed exon 7 of BUD23 gene |
| Genetic reagent (*Mus musculus*) | *Mck*-Cre | *Whitnall et al., 2008* | B6.FVB(129S4)-Tg (Ckmm-cre)5Khn/J | Heart and muscle specific Cre-driver |
| Cell line (*Homo-sapiens*) | A549 | ATCC | CCL-185 | Airway epithelial adenocarcinoma |
| Antibody | anti-BUD23 (Rabbit polyclonal) | GeneTex | Cat# GTX105840 | WB (1:500) |
| Sequence-based reagent | BUD23_F | This paper | Genotyping primers | 5'-CCTGGCTGATGTGTTGCTTT-3' Annealing temperature: 62°C |
| Sequenced-based reagent | BUD23_R | This paper | Genotyping primers | 5'-GCTGCACATCTCTCCTCACT-3' Annealing temperature: 62°C |
| Sequence-based reagent | MT-ND1_F | This paper | qPCR primers | 5'-GAAGTCACCCTAGCCATCATTC-3' |

*Continued on next page*

*Continued*

| Reagent type (species) or resource | Designation | Source or reference | Identifiers | Additional information |
|---|---|---|---|---|
| Sequenced-based reagent | MT-ND1_R | This paper | qPCR primers | 5'GCAGGAGTAATCAGAGGTGTTC-3' |
| Sequenced-based reagent | siRNA: nontargeting control 1 | Thermo Fisher | AM4611 | Silencer Select |
| Sequenced-based reagent | siRNA: nontargeting control 2 | Thermo Fisher | 4390847 | Silencer Select |
| Sequenced-based reagent | siRNA: BUD23 | Thermo Fisher | 41529 | Silencer Select |
| Sequenced-based reagent | siRNA: BUD23 | Thermo Fisher | 41530 | Silencer Select |
| Sequenced-based reagent | siRNA: BUD23 | Thermo Fisher | 41531 | Silencer Select |
| Software, algorithm | R | www.r-project.org | | Key packages:<br>-EDGER<br>-BIOCONDUCTOR<br>-EDASEQ |
| Commercial assay or kit | Seahorse XF Mito stress test | Agilent | 103015–100 | |
| Commercial assay or kit | Seahorse XF Glyco stress test | Agilent | 103020–100 | |

## Animals

All mice were routinely housed in 12:12 light/dark (L:D) cycles with ad libitum access to food and water. All experiments were carried out in accordance with the Animals (Scientific Procedures) Act 1986 (UK) under Home Office protocol number 70/8768 and P3A97F3D1. In studies utilizing conditional targeted mice, littermate controls (floxed/floxed) were used as control; these mice carried no copies of the Cre or iCre. Genotyping was performed on all experimental animals.

## Physiological monitoring

To assess body composition, whole body, fat mass and lean body mass was assessed using the EchoMRI system (Echo Medical Systems). To assess metabolic gas exchange, mice were individually housed in indirect calorimetry cages (CLAMS, Columbus Instruments). Mice were acclimatised to the cages for 48 hr prior to recording data. Measurements of $O_2$ consumption and $CO_2$ production were recorded every 10 min for >72 hr. RER was derived from these measures ($VCO_2/VO_2$), as was energy expenditure ($3.815^*VO_2 + 1.232^*VCO_2$).

## Generation of *Bud23* conditional KO mice

In order to conditionally KO the *Bud23* gene by Cre recombinase expression we generated a transgenic mouse line with the critical exon seven flanked by LoxP sites. Exon seven has an unequal splicing phase, meaning its Cre mediated excision would lead to a frame shift and KO of the *Bud23* gene. This exon is also flanked by large introns giving sufficient space to insert LoxP sites without perturbing normal gene splicing and regulation. We used CRISPR (clustered regularly interspaced short palindromic repeats)-Cas9 in the mouse zygote to integrate these LoxP sites.

Briefly, sgRNA targeting this genomic region, and with minimal off target potential, were identified (sgRNA-1 GGCATTGGGCCTACTTAAAG-GGG, sgRNA-2 AGTTGAAGGGTTCCATAATG-AGG, sgRNA-3 CTTTACAGCCCAAGACCACT-TGG) and selected for use. sgRNA was synthesised in vitro, using protocols developed by *Bassett et al. (2013)*. Forward primers of the form GAAATTAATAC-GACTCACTATA GGN$_{18-20}$GTTTTAGAGCTAGAAATAGC (where GGN$_{18-20}$ is the sgRNA sequence) for each guide were combined with the universal reverse primer, AAAAGCACCGACTCGGTGCCAC TTTTTCAAGTTGATAACGGACTAGCCTTATTTTAACTTGCTATTTCTAGCTCTAAAAC and used in a PCR reaction with high fidelity polymerase (Phusion, NEB). 200 ng of the resulting amplicon were used overnight in an in vitro transcription reaction (HiScribe, NEB, as per manufacturer's instructions), before purification (Megaclear, Ambion) and quantification by nanodrop.

A double stranded DNA (dsDNA) repair template was designed to (i) incorporate loxP sites, with unique restriction sites for screening purposes, 300 bp upstream and 200 bp downstream of exon seven respectively and (ii) integrate silent shield mutations for all three sgRNA to prevent cutting of the repair template and (iii) contain 1000 bp 5' and 3' homology arms to the target region. The above template was synthesised in a pUC57 vector (Genscript, US), the linear 5'homology-loxP-exon7-loxP-3'homology fragment excised from the vector by restriction digest, gel extracted (Bioline) and further cleaned prior to microinjection (Monarch PCR purification kit, NEB).

An injection mix of the three sgRNA (40 ng/ul each), Cas9 mRNA (100 ng/ul) and the dsDNA repair template (10 ng/ul) was prepared and directly microinjected into B6D2F1 (Envigo) zygote pronuclei using standard protocols. Zygotes were cultured overnight and the resulting 2 cell embryos surgically implanted into the oviduct of day 0.5 post-coitum pseudopregnant mice.

Potential founder mice were screened by PCR and digest. Animals positive for both 5' and 3' LoxP integration had the region fully verified by pCR-Blunt cloning followed by Sanger sequencing. two founder mice were identified and then back-crossed to C57BL/6J wild-type mice to assess germline penetrance. Additionally, the described CRISPR targeting strategy resulted in deletion of critical exon 7, a global *Bud23* KO line was generated from this founder animal.

## Genotyping

DNA was extracted from ear snip or tail tip using REDExtract-N-Amp tissue PCR kit (Sigma). Primer sequences and PCR reaction conditions are listed in the key resources table and the primer sequences supplementary file. PCR products were resolved on 1.5% agarose gels.

## Cells

STR authenticated A549 cells were obtained from the European Collection of Cell Cultures (ECACC Cat no: 86012804) and maintained in culture, incubated at 37°C in humidified air, 5% v/v $CO_2$. Cultured cells were tested regularly for mycoplasma contamination. A549 cells were cultured in DMEM growth medium (Sigma, D6546), supplemented with 2 mM L-Glutamine (Sigma, G7513), 10% foetal calf serum (FCS) (Sigma, F96665).

## Depletion of *BUD23* with siRNA

A549 cells were plated at $1 \times 10^6$ cells in a 10 cm cell culture dish and transfected with *BUD23* specific or non-targeting control siRNA (S41529, S41530, S41531, negative control 1, negative control 2, Ambion Silencer Select) using Dharmafect DF-1 according to the manufacturer's guidelines.

## Mass spectrometry

A549 cell pellets were lysed in M-PER mammalian protein extraction reagent (Thermo Scientific). Whole mouse hearts were dissected, quartered and washed in ice cold PBS to remove blood before disruption of the tissue in 1x TBS supplemented with protease inhibitor cocktail using a TissueRuptor (Qiagen). SDS was added to a final concentration of 4% (w/v) prior to sonication. Samples were then reduced with 100 mM DTT and heated for 3 min at 95°C. Samples were prepared for label-free quantification in accordance with published protocols (*Hernandez-Valladares et al., 2016*). Digested samples were analysed by LC-MS/MS using an UltiMate 3000 Rapid Separation LC (RSLC, Dionex Corporation, Sunnyvale, CA) coupled to a Q Exactive HF (Thermo Fisher Scientific, Waltham, MA) mass spectrometer using a stepped normalized collision energy centered at 28.

A549 peptide mixtures were separated using a multistep gradient from 95% A (0.1% FA in water) and 5% B (0.1% FA in acetonitrile) to 7% B at 1 min, 18% B at 58 min, 27% B in 72 min and 60% B at 74 min at 300 nL min$^{-1}$, using a 75 mm x 250 μm i.d. 1.7 μM CSH M-Class C18, analytical column (Waters), for a final run time of 90 min. Cardiac samples were separated using a gradient from 5% B to 7% B at 1 min, 18% B at 81.5 min, 27% B at 102 min and 60% B at 104 min, with a final run time of 120 min. The top 12 precursors were selected for fragmentation automatically by data dependant analysis during each cycle.

Ionization potential was set at 1900V with a survey scan window of 300–1750 m/z. MS1 used a resolution of 120,000 with a target ion intensity of 3e6. Higher-energy collisional dissociation was used. MS2 was set at a resolution of 60,000 with a target ion intensity of 2e5. Maximum injection

times for MS1 and MS2 were 20 ms and 110 ms respectively. Peptides were dynamically excluded for 15 s after one occurrence.

## Mass spectrometry data analysis

Mass spectra were analysed using MaxQuant version 1.6.0.16 (*Cox and Mann, 2008*), searching against either the Uniprot *Mus musculus* database (accessed 21/12/2017) or *Homo sapiens* database (accessed 12/03/2017) using the native Andromeda search engine. Match between runs was selected with a retention time of 1 min, using default parameters for all other settings. MS1 search tolerance was set at 20ppm, and MS2 at 4.5ppm. Modifications were set as Carbamidomethyl (C) for fixed, and Oxidation (M) and Acetyl (Protein N-term) variable. A minimum of 1 spectra was required for identification, while a minimum of 2 spectra were required for quantification. Resulting data was processed using Perseus version 1.6.0.7 (*Tyanova et al., 2016*) and R statistical software (*Wickham, 2009*; *R Development Core Team, 2017*). All proteins identified by site only, as a potential contaminant, or in the decoy reverse database were excluded. Data was then filtered to exclude proteins with more than 3 'zero-values' across all groups. All proteins assigned to the GO term 'blood microparticle' were removed as well. LFQ values were log(2) transformed and missing values imputed in Perseus using a normal distribution (*Lazar et al., 2016*). The width of the new distribution was 0.3 standard deviations of the data, and was shifted downwards by 1.8 standard deviations of the data. A total of 6.68% of values were imputed. Significance was determined using a T-test with a permutation based FDR (FDR < 0.05, s0 = 0.1). s0 represents the relative importance of the difference between the means, with non-zero s0 values taking fold change, and not only p-value, into account. Estimations of copy numbers were performed using the proteomic ruler plug-in version 0.1.6 (*Wiśniewski et al., 2014*). Proteins were then mapped to subcellular localisations to determine percentage protein mass by organelle according to published protocols using the HeLa spatial proteome database (*Doll et al., 2017*; *Itzhak et al., 2016*). Interaction networks were built using String 10 online software (*Szklarczyk et al., 2017*) using seven data sources (textmining, experiments, databases, co-expression, neighborhood, gene fusion co-occurrence) and imported into Cytoscape (*Shannon et al., 2003*). Network edges represent confidence. A minimum interaction score of 0.4 (medium confidence) was required for network inclusion. Enrichment analysis was performed using PANTHER (*Mi et al., 2017*). Lists of proteins were exported for the identified GO terms and then manually coloured using Cytoscape for visualization.

## Echocardiogram

Transthoracic echocardiography (TTE) was performed using a Vevo 770 High-Resolution Imaging System (Visualsonics Inc, Toronto, Canada) and a 30MHz probe. Mice were lightly anesthetized with 1–1.5% isoflurane via facemask, maintaining the heart rate at approximately 450 beats/min. Views were taken in planes that approximated the parasternal short-axis view and the apical long-axis view. M-mode tracings were used to determine left ventricular end-diastolic diameter (LVDD) and end-systolic diameter (LVSD), posterior wall thickness in diastole (LVPWD) and systole (LVPWS), and interventricular septum thickness in diastole (IVSD) and systole (IVSS) over three cardiac cycles. The analysis was performed blinded to animal details. LV fractional shortening (FS) was calculated using the formula FS = [(LVDD - LVSD)/(LVDD)] x 100. Relative wall thickness (RWT) was calculated using the equation RWT = (IVDD + LVPWD)/LVDD.

## Electrocardiogram (ECG)

Mice were lightly anesthetized with 1–1.5% isoflurane via facemask. The body temperature was maintained around 37°C using a heat pad. The lead II ECG was recorded using Power Lab/4SP system (Adinstruments) from needle electrodes inserted subcutaneously into the left and right forelimbs and the right hindlimb. The signal was acquired for about 5 min using Chart 7 software (Adinstruments). During offline analysis, the 5 min recording was examined for unusually shaped P, QRS, or 'T' waves and for time-varying phenomenon (e.g. irregularities in interval durations). Ectopic or abnormal beats are noted. A representative 10–15 s segment of the recording was averaged to obtain the signal averaged ECG to calculate Heart rate, RR interval, P wave duration, PR interval, QRS, JT and QT durations. QT duration was corrected (QTc) using the Bazett's formula (*Bazett, 1997*).

## Polysome profiling

Polysome profiles were prepared according to the protocols previously published in: (*Gandin et al., 2014*). Two conditions were profiled, *BUD23* siRNA and Control siRNA, with an n of 3 samples per condition. Fractions were collected from each sample and RNA was purified using the Trizol extraction method. RNA from fractions containing three or less ribosomes were pooled ('Low translational efficiency) and RNA from fractions containing more than three ribosomes ('high efficiency') were pooled. Sub-polysomal fractions were similarly pooled. RNA sequencing was used to quantify the expression of RNA species in each of the pooled samples. Translational efficiency (TE) was defined as the relative abundance of a transcript in the high and low efficiency pooled fractions. A TE score for each transcript was calculated by averaging the ratio derived from each of the three replicates in each condition. TE was plotted for the two conditions using R software.

Translational efficiency (TE) was then compared to both the length and GC content of the 5' UTR region. 5' UTRs were defined using the TxDb.Hsapiens.UCSC.hg38.knownGene (Bioconductor Core Team and Bioconductor Package Maintainer, 2016) and org.Hs.eg.db (*Carlson, 2018*) R packages. Sequences of the regions were extracted using the twoBitToFa program from the BLAT suite (*Kent, 2002*) and then GC content was calculated using the seqinr R package (*Charif and Lobry, 2007*). The Pearson correlation coefficients, and their relative significances, were calculated using cor.test function from the stats R package (*R Development Core Team, 2017*).

## Assessment of mitochondrial function using seahorse

Live cell metabolic assays were performed on A549 cells pre-treated with either *BUD23* siRNA or control siRNA using a Seahorse XFe 96 analyser (Agilent). Mitochondrial stress tests was performed according to the manufacturer's instructions (Agilent). 2M Oligomycin (OA), 1M FCCP and 0.5M rotenone/antimycin A (AA+R) were used for all conditions. A549 cells were plated into Seahorse XF96 plates at 160,000 cells per well, utilising 16 wells per condition. Cell density was normalised using SRB assay. Experiments were performed in triplicate.

## Assessment of mitochondrial function using oroboros

Of the three distinct experimental preparations available (isolated mitochondria, permeabilized fibers, and tissue homogenates), we utilized cardiac homogenates for mitochondrial assessment (see *Goo et al., 2013* for advantages of this method). Ventricular tissue (~20 mg) was weighed and transferred to 1.5 ml of ice-cold MiRO5 medium (in mM: EGTA 0.5, $MgCl_2$ 1.4, taurine 20, $KH_2P0_4$ 10, HEPES 20, BSA 1%, K-MES 60 mM, sucrose 110 mM, pH 7.1, adjusted with 5 N KOH). Tissue was homogenized for 3 s in 1 s bursts with a tissue homogenizer and loaded immediately (40 µg/ml) into an Oroboros Oxygraph 2 k high resolution respirometry system (Oroboros Instruments, Innsbruck, Austria) for measurement of mitochondrial respiration combined with the Fluorescence-Sensor Green of the O2k-Fluo LED2-Module for $H_2O_2$ measurement (see *Makrecka-Kuka et al., 2015* for full details).

Two identical respiration chambers (chamber A and chamber B) held at the same temperature were run in parallel for each experimental run. All measurements of respiration rates were carried out at 37˚C. Oxygen electrodes were calibrated daily with air-saturated respiration solution Zero calibrations were achieved by injecting yeast into the experimental chambers. Oxygen solubility in the assay medium was calculated as described previously (*Lienig and Forstner, 1984*). $H_2O_2$ flux was measured simultaneously with respiration in the O2k-Fluorometer using the $H_2O_2$-sensitive probe Amplex UltraRed (10 µM) with 1 U/mL horse radish peroxidase (HRP) and 5 U/mL superoxide dismutase (SOD). AmR in the presence of $H_2O_2$ is catalysed by HRP to the fluorescent product resorufin. Calibrations were performed with two sequential injections of $H_2O_2$ at 0.1 µM steps.

Three parameters are commonly used to assess mitochondrial function (for reviews, see *Brand and Nicholls, 2011*; *Pesta and Gnaiger, 2012*). Firstly, the capacity for oxidative phosphorylation (OXPHOS) is the respiratory capacity of mitochondria in the ADP-activated state of oxidative phosphorylation (saturating concentrations of ADP, inorganic phosphate, oxygen, and defined substrates). Secondly, LEAK respiration rate represents mitochondrial respiration that occurs in the absence of ATP generation, mainly to compensate for proton leak across the mitochondrial inner membrane. Lastly, the respiratory electron transfer-pathway capacity (ETC) is mitochondrial respiration in the noncoupled state in the presence an uncoupler; this induces maximum oxygen flux

through the electron transport chain. The Respiratory Control Ratio (RCR, calculated here as OXPHOS/LEAK) provides a measure of the degree of coupling between oxidation and phosphorylation, or in other words, the efficiency of mitochondrial ATP production.

OXPHOS, LEAK and ETS were measured in the presence of Complex I substrates pyruvate and malate (electron transfer through Complexes I-IV) or Complex I+II substrates (addition of succinate). Additionally, respiratory flux with electron transfer through Complex IV alone was measured via the addition of the electron donor tetramethyl-phenylene-diamine (TMPD). The protocol used to measure these parameters was adapted from *Pesta and Gnaiger (2012)*. Briefly, pyruvate (5 mmol l$^{-1}$), malate (0.25 mmol l$^{-1}$) and glutamate (10 mmol l$^{-1}$) are added as carbon substrates and to spark the citric acid cycle. Under these conditions, mitochondria are in LEAK respiration with CI substrates in the absence of adenylates. OXPHOS with CI substrates was achieved through addition of saturating levels of ADP (2 mM l$^{-1}$). Following steady-state conditions, succinate (10 mmol l$^{-1}$) was added to achieve OXPHOS with CI+CII substrates. To uncouple respiration and achieve ETS with CI+CII substrates, carbonyl cyanide 4-trifluoromethoxyphenylhydrazone (FCCP) was carefully titrated to a maximum concentration of 0.25 µmol l$^{-1}$. Rotenone was then added to achieve ETS with CII substrates and antimycin A (5 µmol l$^{-1}$) was given to block Complex III and measure background non-mitochondrial residual oxygen consumption (ROX). OXPHOS through Complex IV alone was assessed by adding the electron donor TMPD (0.5 mmol l$^{-1}$). To avoid oxidation of TMPD, ascorbate (2 mMol l$^{-1}$) was added prior to TMPD injection.

## Citrate synthase assay

The citrate synthase activity of cardiac homogenates were measured in frozen homogenates. Briefly, maximal activity (Vmax) was determined with a spectrophotometer at 37°C with a Synergy HTX spectrophotometer (BioTek, UK) in assay buffer; 50 mM TRIS-HCl, pH 8.0. Citrate synthase activity was monitored in the presence or absence of oxaloacetate by the appearance of 5-thio-2-nitrobenzoic acid as a result of the reaction of free acetyl-CoA with 5,5'-dithiobis(2-nitrobenzoic acid) at 412 nm over a 10 min incubation period (assay buffer with 0.5 oxaloacetate, 0.3 mM acetyl-CoA, 0.15 mM 5,5-dithiobis-2-nitrobenzoic acid). Extinction coefficients were empirically determined to quantify Vmax values. Enzyme activities were normalised to total soluble protein, which was quantified according to Bradford (*Bradford, 1976*).

## Western blots

Total cell protein was isolated from cells using protein extraction buffer (50 mM Tris, 150 mM NaCl, 1% TritonX-100, supplemented with protease inhibitors). Total protein was isolated from snap-frozen tissue using RIPA buffer supplemented with protease inhibitor cocktail after disruption using Lysing Matrix D tubes (MP Bio).

## Electron Microscopy

All experiments were performed according to current UK Home Office regulations and under approval of the relevant University of Manchester (Manchester, UK) local ethics committee.

Six mice were used for this study (n = 3 control, n = 3 Bud 23 KO). Immediately after euthanasia, hearts were harvested and small portions (cubes with sides smaller than 0.5 mm) of right and left ventricle were immersion fixed in 2.5% glutaraldehyde and 2% paraformaldehyde in 100 mM sodium cacodylate buffer (pH 7.2) overnight. Specimen preparation was performed, with small modifications, according to the Ellisman protocol (*Holcomb et al., 2013*). Briefly, after washings in sodium cacodylate, samples were sequentially stained in: 2% osmium tetroxide and 1.5% potassium ferrocyanide in 100 mM sodium cacodylate for 1 hr; 1% aqueous thiocarbohydrazide for 20 min; 2% aqueous osmium for 30 min; 1% aqueous uranyl acetate overnight and Walton's lead aspartate for 30 min the following morning at 60°C. Samples were washed 3 times for 10 min in double distilled water after each staining step. Staining was followed by dehydration in ethanol ascending series (50%, 70%, 90%, 100%) and infiltration with TAAB 812 hard resin mixed with propylene oxide. After embedding, resin blocks were cured at 70°C for 40 hr. Plastic blocks were sectioned at 80 nm thickness using a Leica UC6 ultramicrotome and sections were imaged using a FEI Tecnai12 Biotwin operated at 100kV. Images were analysed using Fiji (*Schindelin et al., 2012*).

## Cell fractionation

Fractionation experiments were performed by differential centrifugation as previously described previously (*Rorbach et al., 2014*).

## Data analysis and statistics

Unless otherwise stated in the figure legend, parametric data was analysed by ANOVA with a Dunnett's multiple comparisons test. Non-parametric data was analysed using a Mann-Whitney test. RNAseq data was analysed using edgeR (*Robinson et al., 2010*). For Mass Spectrometry data analysis please refer to the dedicated section.

# Acknowledgements

We would like to acknowledge the University of Manchester Genomics technology facility; Michael Smiga, Leo Zeef, and Andy Hayes. The Biological Mass Spectrometry Core Research Facility for technical support. Carolyn Jones for technical support with polysome profiling. The University of Manchester electron microscopy core facility. DWR and ASL are Wellcome Investigators, Wellcome Trust (107849/Z/15/Z, 107851/Z/15/2).

# Additional information

### Funding

| Funder | Grant reference number | Author |
|---|---|---|
| Medical Research Council | MR/L010240/1 | David Ray |

The funders had no role in study design, data collection and interpretation, or the decision to submit the work for publication.

### Author contributions

Matthew Baxter, Conceptualization, Data curation, Formal analysis, Investigation, Visualization, Methodology, Writing - original draft, Writing - review and editing; Maria Voronkov, Data curation, Formal analysis, Investigation, Visualization, Methodology, Writing - original draft, Writing - review and editing; Toryn Poolman, Conceptualization, Data curation, Formal analysis, Methodology, Writing - review and editing; Gina Galli, Data curation, Formal analysis, Methodology, Writing - original draft; Christian Pinali, Data curation, Formal analysis, Methodology; Laurence Goosey, Data curation, Investigation; Abigail Knight, Karolina Krakowiak, Investigation; Robert Maidstone, Mudassar Iqbal, Formal analysis, Methodology; Min Zi, Formal analysis, Investigation, Methodology, Writing - original draft; Sukhpal Prehar, Formal analysis, Investigation, Methodology; Elizabeth J Cartwright, Formal analysis, Supervision, Methodology; Julie Gibbs, Conceptualization, Resources, Investigation, Project administration; Laura C Matthews, Conceptualization, Investigation; Antony D Adamson, Neil E Humphreys, Resources, Investigation, Methodology; Pedro Rebelo-Guiomar, Investigation, Methodology; Michal Minczuk, Supervision, Methodology; David A Bechtold, Supervision, Writing - original draft; Andrew Loudon, Conceptualization, Supervision, Funding acquisition, Writing - review and editing; David Ray, Conceptualization, Supervision, Funding acquisition, Writing - original draft, Project administration, Writing - review and editing

### Author ORCIDs

Matthew Baxter (iD) https://orcid.org/0000-0002-3612-2574
Maria Voronkov (iD) https://orcid.org/0000-0001-5636-9892
Michal Minczuk (iD) http://orcid.org/0000-0001-8242-1420

### Ethics

Animal experimentation: All experiments were carried out in strict accordance with the Animals (Scientific Procedures) Act 1986 (UK) and protocols were approved by an internal ethics committee at

the University of Manchester. Every effort was made to minimize suffering. Home office project licence 70/8768 and P3A97F3D1.

## Decision letter and Author response
Decision letter https://doi.org/10.7554/eLife.50705.sa1
Author response https://doi.org/10.7554/eLife.50705.sa2

# Additional files

## Supplementary files
• Supplementary file 1. Primer sequences.

## Data availability
RNAseq data have been deposited to ArrayExpress, under the accession code E-MTAB-8673. All proteomics data is included in the supporting files, and raw data have been deposited to PRIDE under the accession code PXD017019.

The following datasets were generated:

| Author(s) | Year | Dataset title | Dataset URL | Database and Identifier |
|---|---|---|---|---|
| Maria Voronkov, Toryn Poolman, Gina Galli, Christian Pinali, Laurence Goosey, Abigail Knight, Karolina Krakowiak, Robert Maidstone, Mudassar Iqbal, Min Zi, Sukhpal Prehar, Elizabeth J Cartwright, Julie Gibbs, Laura C Matthews, Antony D Adamson, Neil E Humphreys, Pedro Rebelo-Guiomar, Michal Minczuk, David A Bechtold, Andrew Loudon, David Ray, Matthew Baxter | 2020 | Data from: Cardiac mitochondrial function depends on BUD23 mediated ribosome programming | https://www.ebi.ac.uk/arrayexpress/experiments/E-MTAB-8673/ | ArrayExpress, E-MTAB-8673 |
| Maria Voronkov, Toryn Poolman, Gina Galli, Christian Pinali, Laurence Goosey, Abigail Knight, Karolina Krakowiak, Robert Maidstone, Mudassar Iqbal, Min Zi, Sukhpal Prehar, Elizabeth J Cartwright, Julie Gibbs, Laura C Matthews, Antony D Adamson, Neil E Humphreys, Pedro Rebelo-Guiomar, Michal Minczuk, David A Bechtold, Andrew Loudon, David Ray, Matthew Baxter | 2020 | The role of Bud23 in mouse heart tissue | https://www.ebi.ac.uk/pride/archive/projects/PXD017019 | PRIDE, PXD017019 |

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
