## [Decision Letter]

**Acceptance summary:**

Given the essential role of mitochondria for cells, the regulation of expression of mitochondrial-destined proteins encoded by mRNAs that are translated at cytosolic ribosomes is a topic of general interest. The authors describe a previously unknown physiological function of the ribosome biogenesis factor, BUD23, in the expression of mitochondrial proteins at the level of translation. Comprehensive transcriptomic and proteomic analysis in cellular and mouse models delineates BUD23 potential role in human disease connected with cardiomyopathy. The author's work initiates further fundamental research on other factors associated with ribosome biogenesis that can influence the selective production of mitochondrial proteins in the cytosol.

**Decision letter after peer review:**

Thank you for submitting your article "Cardiac mitochondrial function depends on BUD23 mediated ribosome programming" for consideration by *eLife*. Your article has been reviewed by four peer reviewers and the evaluation has been overseen by Didier Stainier as the Senior Editor. The reviewers have opted to remain anonymous.

The reviewers have discussed the reviews with one another and the Reviewing Editor has drafted this decision to help you prepare a revised submission.

Summary:

Baxter et al. seek to elucidate how BUD23, an rRNA methyltransferase involved in ribosome maturation, can affect mitochondrial content and function. Regulation of mitochondrial gene expression is highly complex and the identification of nuclear-encoded genes that exert levels of control represents an important contribution to our understanding. The authors provide a broad analysis of changes in mRNA and protein levels upon BUD23 deficiency in mammalian cells and mouse models. The authors conclude that BUD23 is critical for the expression of genes involved in oxidative phosphorylation and general mitochondrial function, possibly by its role in selectively promoting translation mRNAs with low 5' UTR GC content. They also found that BUD23 double knockout in mice is embryonic lethal, and haploinsufficiency affects fitness during development. Specifically, they found that disrupting BUD23 expression in cardiomyocytes caused gross heart defect, which leads to lethality in juvenile mice.

Overall, the manuscript is interesting and comprehensive and will add more weight to the growing complexity of post-transcriptional regulation at the level of translation. However, the reviewers' general concern is to what extent the observed decrease in translation efficiency of selected targets is specific to BUD23 depletion or part of a more general mechanism. To address this issue we have the following suggestions to improve the manuscript.

Essential revisions:

1) It remains unclear whether the decreased translation efficiency of mRNA of nuclear-encoded mitochondrial proteins is specific to the loss of BUD23 or part of a general mechanism resulting from 40S/60S imbalance. Ideally, to address this question another important ribosome biogenesis factor should be tested performing carefully selected experiments. However, as a minimum, the authors should perform an experiment to check if 18SE pre-rRNA is still incorporated in mature 40S and even translating polysomes. This could help reinforce author’s claim of BUD23 role in mitochondrial related gene expression through ribosome assembly. Gel pictures that are used to generate Figure 1C ought to be shown, ideally also with northern against pre-RNA for 18S to show for processing defect, especially given that in Haag et al., 2014, there are differences on 18SE pre-rRNA processing in different cell types upon BUD23 knockdown. Further, the authors should consider mitigating their assumption that BUD23 is solely responsible for the observed effect on translation efficiency and should discuss the possibility of the contribution of a more general mechanism resulting from 40S/60S imbalance.

2) To analyse levels of transcripts the authors split the polysome fractions in two or more than two ribosomes bound to mRNAs. The number of ribosomes present on an mRNA depends among other causes, on its length and hence, one can expect that short mRNAs (such as those encoding for many ribosomal proteins) will be bound to fewer ribosomes than longer ones. Hence, upon a global decrease in functional 80S due to BUD23 inactivation, mRNAs of short to intermediate lengths are more likely to shift from the heavy to the light fraction due to the loss of 1 or 2 ribosomes. The authors need to clarify whether the observed decrease in translation efficiency is indeed specific to a subset of mRNAs or if it is a general phenomenon. For instance, this could be addressed by comparing the quantity of some well-chosen mRNAs (i.e. whose translation efficiency is either proposed to be decreased according to the author's analysis or not affected) in each fraction of the polysome profiles by qPCR for cells treated with control or BUD23 siRNAs.

Furthermore, is there any agreement between the transcripts/proteins identified as having lower translation efficiency? Linking proteomics with polysome sequencing might shed more light on possible differential protein stability.

3) The finding that the 5'UTR GC content can define translation efficiency depending on BUD23 is interesting and mechanistically important. However, there is very little follow up on the statement that low translation efficiency correlates with low 5’UTR GC content. Especially there is no link between 5’UTR GC content and translation efficiency of mitochondrial related transcripts. Some questions to consider by the authors; Is translation efficiency reduction for low 5'UTR GC content in BUD23 siRNA treated cells mainly due to mitochondrial-encoded and other mitochondrial genes or is it a more general set of genes that so happen to be enriched for mitochondrial related genes? How about other cytosolic proteins that are downregulated in the proteomics analysis? Do they have low 5'UTR% GC content too? Trying to see if a high significance of low translation efficiency in Figure 1J mostly comes from mitochondrial related genes or low% GC content at 5'UTR.

---

## [Author Response]

Essential revisions:1) It remains unclear whether the decreased translation efficiency of mRNA of nuclear-encoded mitochondrial proteins is specific to the loss of BUD23 or part of a general mechanism resulting from 40S/60S imbalance. Ideally, to address this question another important ribosome biogenesis factor should be tested performing carefully selected experiments. However, as a minimum, the authors should perform an experiment to check if 18SE pre-rRNA is still incorporated in mature 40S and even translating polysomes. This could help reinforce author’s claim of BUD23 role in mitochondrial related gene expression through ribosome assembly. Gel pictures that are used to generate Figure 1C ought to be shown, ideally also with northern against pre-RNA for 18S to show for processing defect, especially given that in Haag et al., 2014, there are differences on 18SE pre-rRNA processing in different cell types upon BUD23 knockdown. Further, the authors should consider mitigating their assumption that BUD23 is solely responsible for the observed effect on translation efficiency and should discuss the possibility of the contribution of a more general mechanism resulting from 40S/60S imbalance.

The reviewers raise an important, and an interesting question: whether the observed effects on mitochondrial content and function we have reported are specific to the loss of BUD23, or part of a more general mechanism resulting from 40S/60S imbalance. In order to address this, we have taken the reviewers suggestion of testing other important ribosome biogenesis factors (LTV1 and RIOK2) as well as the ribosomal small subunit protein RPS27A, which is also reported to result in ribosomal subunit imbalance. Transient siRNA-mediated knockdown of ribosome small subunit biogenesis factors LTV1, RIOK2 and RPS27A all resulted in significant decrease in 18S/28S ratio relative to control siRNA, indicating a similar 40S/60S imbalance as is seen with BUD23 knockdown (Figure 2G, H, I). Knockdown of BUD23 resulted in a significant decrease in mitochondrial transcript expression, as was observed in the original manuscript. LTV1 knockdown also resulted in a similar decrease in mitochondrial transcript expression. However RPS27A and RIOK2 knockdown did not.

When mitochondrial copy number was measured a small decrease was found in the BUD23 knockdown group, no effect in the RPS27A knockdown group, and small increase in the LTV1 and RIOK2 groups. It is particularly notable that the effects on mitochondrial content and transcript expression do not correlate with the reduction in 18S/28S ratio. When taken together, we interpret these data to indicate that disruption of 40S biogenesis is not by itself sufficient to result in mitochondrial dysfunction. The observation that LTV1 knockdown results in similar decreases to levels of mitochondrial transcript abundance as BUD23 knockdown is interesting, and indicates that there is a biogenesis factor specific impact on the ribosome, and its capacity to provide proteins for efficient mitochondrial function. It may be of interest in the future to undertake further in-depth studies of various ribosomal biogenesis factors and their impact on the function of the final ribosome, and the resulting impact on mitochondrial function. These new data have been included into the revised manuscript in Figure 2, and we discuss in the paper the BUD23 specific impact on ribosome regulation of mitochondrial function.

Example gel pictures used to quantify 18S/28S ratio have also been added to the manuscript in Figure 2—figure supplement 2D.

Lastly, we have mitigated our manuscript Discussion point that BUD23 is solely responsible for the observed effect on translation. As shown in the new data (Figure 2G, H, I), 40S/60S imbalance does not always lead to mitochondrial disruption, however we do observe that ribosome biogenesis factors other than BUD23 (e.g. LTV1) may result in similar phenotypes.

2) To analyse levels of transcripts the authors split the polysome fractions in two or more than two ribosomes bound to mRNAs. The number of ribosomes present on an mRNA depends among other causes, on its length and hence, one can expect that short mRNAs (such as those encoding for many ribosomal proteins) will be bound to fewer ribosomes than longer ones. Hence, upon a global decrease in functional 80S due to BUD23 inactivation, mRNAs of short to intermediate lengths are more likely to shift from the heavy to the light fraction due to the loss of 1 or 2 ribosomes. The authors need to clarify whether the observed decrease in translation efficiency is indeed specific to a subset of mRNAs or if it is a general phenomenon. For instance, this could be addressed by comparing the quantity of some well-chosen mRNAs (i.e. whose translation efficiency is either proposed to be decreased according to the author's analysis or not affected) in each fraction of the polysome profiles by qPCR for cells treated with control or BUD23 siRNAs.Furthermore, is there any agreement between the transcripts/proteins identified as having lower translation efficiency? Linking proteomics with polysome sequencing might shed more light on possible differential protein stability.

In order to address the possibility of shorter transcripts being more likely than longer transcripts to shift from the heavy to the light fraction we have plotted the transcript length against the change in translational efficiency (TE), and performed a regression analysis.

As shown in Figure 1—figure supplement 3, shorter transcripts were *not*more likely than longer transcripts to exhibit a decrease in translational efficiency in response to loss of BUD23. There is in fact a weak correlation in the opposite direction. These additional observations are now referenced in the revised manuscript text.

We conclude that the observed decrease in the translational efficiency is indeed specific to a subset of transcripts: first of all, not all transcripts showed a decrease in TE. The majority of transcripts showed little or no change in TE, and some transcripts showed an increase in TE (Figure 1F). Secondly, we analysed global translational rates using 35S-methionine incorporation assay which showed no detectable change in overall protein production in this model (Figure 1—figure supplement 1A). The strongest predictor of a change in translational efficiency remained the 5’UTR GC content.

To measure the association between the change in mRNA TE and encoded protein abundance, we performed a correlation analysis between the polysome profiling data and the mouse heart proteomics (Figure 4—figure supplement 2). Although there is a general trend of correlation between change in TE and change in protein abundance with BUD23 deficiency, the correlation is extremely weak. We believe this observation to be unsurprising for a number of reasons, especially the fact that the analyses were completed in two completely different models (TE was measured in a human cell line after transient BUD23 knockdown, whereas the proteomics dataset is in 26-day old mouse heart tissue). Furthermore, the two methods (polysome profiling versus proteomics) have fundamental differences in data generation which may make correlation incompatible – for example whilst we were able to detect transcripts from over 14000 different genes in the polysome profiling, we were only able to confidently detect around 3000 proteins by proteomics. Furthermore, TE is a relative score derived from ratios of transcripts in heavy and light polysome fractions, rather than an absolute quantification of abundance, such as is achieved by proteomics. We have included this extra correlation analysis in the updated manuscript, however we are reluctant to read too deeply into correlations between TE and proteomics, as further biological mechanisms are at play here, not the least of which is protein stability. We believe it is of note, however, that the mitochondrial phenotype was conserved between an in vitrohuman cell model, and an in vivo murine heart model, and we believe this conservation between models increases the confidence that our observations are reflective of an important biological control circuit.

3) The finding that the 5'UTR GC content can define translation efficiency depending on BUD23 is interesting and mechanistically important. However, there is very little follow up on the statement that low translation efficiency correlates with low 5’UTR GC content. Especially there is no link between 5’UTR GC content and translation efficiency of mitochondrial related transcripts. Some questions to consider by the authors; Is translation efficiency reduction for low 5'UTR GC content in BUD23 siRNA treated cells mainly due to mitochondrial-encoded and other mitochondrial genes or is it a more general set of genes that so happen to be enriched for mitochondrial related genes? How about other cytosolic proteins that are downregulated in the proteomics analysis? Do they have low 5'UTR% GC content too? Trying to see if a high significance of low translation efficiency in Figure 1J mostly comes from mitochondrial related genes or low% GC content at 5'UTR.

This is an interesting and important question. In order to address this we performed gene ontology (GO) analysis on the transcripts with the strongest down-regulation of translation efficiency (Figure 1—figure supplement 1B). There were many GO terms which were significantly over-represented in this analysis – notably a number of terms linked to “metabolic processes” as well as “organelle organization” and “protein transport”. These genes could contribute to the observed mitochondrial phenotype. However there were also groups of genes in the analysis that are not obviously linked to mitochondria. In a further analysis we have examined the 5’UTR GC content of all the detected mitochondrial transcripts compared to all the transcripts in the analysis (Figure 1—figure supplement 3B). This new analysis shows that there is no significant difference between distribution of 5’UTR GC content in mitochondrial related transcripts relative to all transcripts. This indicates that the correlation between low 5’UTR GC content and decreased mRNA TE is a general phenomenon, applying to all transcripts, rather than an artefact of over-represented mitochondrial transcripts. We therefore conclude that the subsequent mitochondrial phenotypes are likely due to reduced translation of a subset of mitochondrial and mitochondria-regulating transcripts, including such genes as mTOR. This extra analysis is now discussed in the manuscript.